# Learning the Structure of Large Networked Systems Obeying Conservation Laws[*]

**Anirudh Rayas**
Arizona State University
ahrayas@asu.edu

**Rajasekhar Anguluri**
Arizona State University
rangulur@asu.edu

**Gautam Dasarathy**
Arizona State University
gautamd@asu.edu

## Abstract

Many networked systems such as electric networks, the brain, and social networks of opinion dynamics are known to obey conservation laws. Examples of this phenomenon include the Kirchoff laws in electric networks and opinion consensus in social networks. Conservation laws in networked systems are modeled as *balance equations* of the form $X = B^*Y$, where the sparsity pattern of $B^* \in \mathbb{R}^{p \times p}$ captures the connectivity of the network on $p$ nodes, and $Y, X \in \mathbb{R}^p$ are vectors of *potentials* and *injected flows* at the nodes respectively. The node potentials $Y$ cause flows across edges which aim to balance out the potential difference, and the flows $X$ injected at the nodes are extraneous to the network dynamics. In several practical systems, the network structure is often unknown and needs to be estimated from data to facilitate modeling, management, and control. To this end, one has access to samples of the node potentials $Y$, but only the statistics of the node injections $X$. Motivated by this important problem, we study the estimation of the sparsity structure of the matrix $B^*$ from $n$ samples of $Y$ under the assumption that the node injections $X$ follow a Gaussian distribution with a known covariance $\Sigma_X$. We propose a new $\ell_1$-regularized maximum likelihood estimator for tackling this problem in the high-dimensional regime where the size of the network may be vastly larger than the number of samples $n$. We show that this optimization problem is convex in the objective and admits a unique solution. Under a new mutual incoherence condition, we establish sufficient conditions on the triple $(n, p, d)$ for which exact sparsity recovery of $B^*$ is possible with high probability; and $d$ is the degree of the underlying graph. We also establish guarantees for the recovery of $B^*$ in the element-wise maximum, Frobenius, and operator norms. Finally, we complement our theoretical results with experimental validation of the performance of the proposed estimator on synthetic and real-world data.

## 1 Introduction

Let $\mathcal{G} = ([p], E)$ be a directed graph on the vertex set $[p] \triangleq \{1, 2, \ldots, p\}$ with a size $m$ edge set $E \subset [p] \times [p]$. Let $D$ denote the $p \times m$ incidence matrix that encodes the edges of $\mathcal{G}$ as follows: each column of $D$ is associated with an edge $(i, j) \in E$ and is a vector of zeros except at the locations $i$ and $j$ where it is $-1$ and $+1$ respectively. Let $X \in \mathbb{R}^p$ be a vector of *injected flows or signals* at the vertices and let $f \in \mathbb{R}^m$ be the vector of *flows* through the edges. Then, the basic *conservation law* between these flows may be expressed as $Df + X = 0$; that is, at each vertex, the flow (which is a linear combination of flows at the edges incident on the vertex) has to balance the injections. In physical systems, edge flows $f$ often arise as a way to balance the differences between certain *potentials* $Y \in \mathbb{R}^p$ at the vertices. That is, the flows satisfy $f = -D^\mathsf{T} Y$; notice that this implies that

---
[*]This work was supported in part by the National Science Foundation (NSF) under the grants CCF-2048223 and OAC-1934766, and by the National Institutes of Health (NIH) under the grant 1R01GM140468-01.

the flow at the edge $(i, j)$ is given by $Y_j - Y_i$. Thus, the above conservation law yields the following relationship, called a *balance equation*, between the node potentials and injected flows:

$$X - B^* Y = 0, \tag{1}$$

where $B^* \triangleq DD^{\mathsf{T}} \in \mathbb{R}^{p \times p}$ is the symmetric Laplacian matrix [4, 60]. In an electrical circuit (with unit resistances on edges), $Y$ corresponds to the voltage potentials at the vertices, $f$ corresponds to the edge currents, and $X$ denotes the injected currents at the vertices. Indeed, this picture can be generalized by assigning weights to the edges of the network (conductances in the case of an electric network), and allowing the flows to be weighted by these weights. The model in (1) is referred to as generalized Kirchoff's law, and importantly, this models the relationship between flows and potentials in a range of systems that satisfy conservation laws such as hydraulic networks, opinion consensus in social networks, and transportation/distribution networks (see [60, 40, 55] and references therein).

It can be readily seen that the Laplacian $B^*$ is a positive semi-definite that encodes the edges of $\mathcal{G}$. Specifically, $(i, j) \in E$ if and only if $B_{ij}^* \neq 0$. The Laplacian lies at the heart of spectral graph theory [15], and owing to its fascinating properties has found a range of applications in diverse areas such as image processing, manifold learning, spectral clustering, and bandits [52, 6, 62, 58, 32, 56]. In this paper, we consider a situation where the edge set $E$ of the graph is unknown and needs to be estimated from measurements of the node potentials $Y$. Based on the above discussion, we will cast this as a problem of learning an unknown positive definite $B^*$ (or the sparsity pattern thereof) from measurements of $Y$. Further, we suppose that we only have access to the statistics of $X$, namely, that it is a 0-mean Gaussian random vector with a covariance matrix $\Sigma_X$. The situations where $\Sigma_X$ is unknown and $B^*$ is non-invertible is briefly discussed in the remarks in Section 2. We list a variety of applications where this learning problem arises naturally.

1. *Topology learning in electric networks*: Consider an electric network (or circuit) with $p$ nodes, current injections $X$, node voltages $Y$, conductances $A_{ij} \geq 0$ between nodes $i$ and $j$, and shunt conductances $A_{ii} \geq 0$ connecting $i$-th node to the ground. The current-balance equation is given by (1), where $B^*$ is the Laplacian with $B_{ij}^* = -A_{ij}$ and $B_{ii}^* = A_{ii} + \sum_{j=1}^n A_{ij}$ [21]. In real electric grids, current injections are unknown random variables. To ensure reliable power supply, learning $B^*$ and its underlying graph from voltage samples is important and has been widely studied [20, 33, 13, 1]. The current-balance equation also appears in Markov chains and flow networks where Kirchoff laws apply [60, 46].

2. *Brain connectivity from graph filters*: The structural connectivity of the human brain is often studied using a network with nodes representing brain regions, and the edge weights representing the density of anatomical connections [45, 25, 61]. Recent studies showed that the weights can be inferred using *graph filters* satisfying (1) with $B^* = (\sum_{l=0}^{L-1} h_l A^l)^{-1}$, where $A$ is the symmetric adjacency matrix; $h_l$ is the filter coefficient; and $X$ is the latent graph signal. For brain networks, [51, 36] showed that $L = 3$ is a reasonable choice. Additionally, for large $L$, the graph filter $B^*$ can be approximated as $B^* = 1 - \alpha A$ using matrix power series expansion, thus a feasible choice for analysis. Graph filters are also used in social and protein interaction networks [50, 37].

3. *Structural equation models (SEM)*: Structural equation models are used to explain relationships among exploratory variables in several domains; for e.g., psychoanalysis [22], social sciences [16], medical research, and neuroimaging [7, 47, 38]. Using SEMs, [42] provided a causal interpretation of Linear Hawkes Processes. In SEM with no latent variables, we let $y = Ay + x$, where $A$ is the *path matrix*. Then the SEM satisifies (1) with $B^* = I - A$.

4. *Linear dynamical (diffusion) networks*: These network dynamics are satisfied by many systems including consensus dynamics, thermal capacitance networks, power swing dynamics, [54]. Further, by lifting approach, these dynamics can be used to study periodic/cyclic behavior in atmospheric systems [53].

Before we detail our topology discovery method, we comment on a few competing approaches that only have limited utility in our setting. First, penalized (nodal) regression methods [39] are not applicable here since these require samples of both $X$ and $Y$. Yet other works have considered the estimation of topology from linear measurements [2, 9, 33, 65, 11, 14, 18, 19], but these assume direct linear access to the graph (via the adjacency or appropriate covariances) or assume low-rankedness in the underlying structure. A recent line of work [20, 1], proposed estimating $B^*$ by estimating the

inverse covariance (or precision) matrix $\Theta^*$ of $Y$ using the graphical LASSO (GLASSO) [66, 23]. In particular, [20, 1] showed that $\Theta^*$ has non-zeros corresponding to those pairs of vertices that are connected by paths of length at most two; that is, the $(i, j)$-th entry of $\Theta^*$ is non zero if and only if $(i, j)$ is an edge in $\mathcal{G}$ or there is a $k \in [p]$ such that $i - k - j$ is a path in $\mathcal{G}$. The authors then estimated edges of $\mathcal{G}$ by identifying (and eliminating) the pairs of vertices that have two-hop connections in $\mathcal{G}$ (see Fig. 1)—for future reference, we call this estimator as GLASSO+2HR (2 hop refinement). However, this estimator requires strong structural assumptions on $\mathcal{G}$ such as triangle-freeness. Further, the precision matrix $\Theta^*$ of $Y$ is far more dense than the underlying graph $\mathcal{G}$ since $\Theta^* = B^* \Sigma_X^{-1} (B^*)^{\mathsf{T}}$; this results in sub-optimal data requirements for reliable recovery (see Remark 3). Finally, if $\Sigma_X$ is a diagonal matrix, we can estimate the sparsity pattern of $B^*$ by taking the principal square root of the empirical covariance matrix of $Y$. Unfortunately, this method does not allow for any correlation between the node injections which is not the case in practice. Importantly, this method is numerically unstable unless one has a large number of samples $(n)$ of $Y$ so that the empirical covariance matrix is invertible—-a requirement that is at odds with the high-dimensional regime where one typically desires $n$ to be smaller than the number of variables $p$.

In light of the limitations of previous approaches, we study a natural penalized maximum likelihood estimator for $B^*$ using the samples $\{Y_i\}_{i=1}^n$. This estimator is not only statistically efficient but also obviates the restrictive assumptions imposed by the aforementioned methods. We now summarize the main contributions of the paper:

- We propose a novel $\ell_1$ regularized maximum likelihood estimator (MLE) for $B^*$ from samples of $Y$. It is worth noting that the optimization program we propose is not the standard graphical LASSO program [66, 23] as it involves terms that are quadratic in the optimization variable. Our first result shows that, notwithstanding its form, the $\ell_1$ regularized MLE is convex in $B$ and it has a unique minimum even in the high-dimensional regime ($n \ll p$) under certain standard conditions.

- Under a new mutual incoherence condition, we provide a sufficient condition on the number of samples $n$ required to recover the exact sparsity of $B^*$ with high probability. Furthermore, under these sufficient conditions we also establish the consistency of our estimator in the element-wise maximum, Frobenius, and spectral norms. Formally, we show that if $n = \Omega(d^2 \log p)$ then with high probability $\|\widehat{B} - B^*\|_\infty \in \mathcal{O}(\sqrt{\log p / n})$, where $d$ is the degree of the underlying graph.

- Finally, we complement our theoretical results with experimental results both on the synthetic data sets and data from a benchmark power distribution system. Our experiments demonstrate the clear benefit of the proposed estimator over baseline and competing methods.

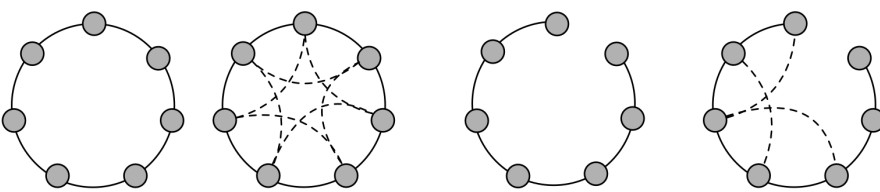

$(a)$ true graph     $(b)$ graphical model     $(c)\, \ell_1 - \text{MLE}$ (this paper)     $(d)$ GLASSO+2HR

Figure 1: Stylistic visualization of $\ell_1$-MLE vs GLASSO+2HR in [20]. (a) True graph $\mathcal{G}$ with $B^*$; (b) graphical model of $\Theta^* = B^* \Sigma_X^{-1} (B^*)^{\mathsf{T}}$; (c) estimate of $B^*$ by our proposed $\ell_1-$regularized MLE estimator; and (d) estimate of $B^*$ by GLASSO+2HR [20]. Graph $\mathcal{G}$ and the graphical model of $\Theta^*$ have same set of vertices; however, in the latter, there are spurious edges (dashed lines in (b)) between vertices that are two-hop neighbors in $\mathcal{G}$ (see main text in Introduction). In Fig 1(c) and 1(d), we depict potential outputs of $\ell_1-$MLE and GLASSO+2HR respectively for some sample covariance matrix. Since the outputs are random objects, the graphs in Fig 1(c) and 1(d) can have missing and false edges that are not present in the true graph in 1(a). The GLASSO+2HR estimate advocated in [20] recovers $B^*$ by first estimating $\Theta^* = B^* \Sigma_X^{-1} (B^*)^{\mathsf{T}}$ which is potentially dense when compared to $B^*$ (see Fig 1(a) and Fig 1(b)). Hence, the GLASSO+2HR estimate in Fig 1(d) might have more edges than the true graph. However, since our $\ell_1$-MLE directly estimates the sparse matrix $B^*$, likely, the estimated and the true graph might only differ on a few edges, as shown in Fig 1 (c). Our experimental results in Section 4 validate this claim.

**Organization of the paper**: In Section 2, we introduce an $\ell_1$-regularized ML estimation problem for networked systems obeying conservation laws. In Section 3, we show that this optimization problem is convex in the objective and establish consistency and support recovery rates for our estimator. In Section 4, we provide simulation results. In Section 5, we summarize our paper with future directions.

**Notation**: For any two subsets $T_1$ and $T_2$ of $[p] \times [p]$, we denote by $A_{T_1 T_2}$ the submatrix of $A$ with rows and columns indexed by $T_1$ and $T_2$, respectively. When $T_1 = T_2$ we denote the submatrix by $A_{T_1}$. For a matrix $A = (A_{i,j}) \in \mathbb{R}^{p \times p}$, we use $\|A\|_\infty \triangleq \max_{i,j} |A_{ij}|$ to denote the maximum element-wise norm, and $\|A\|_F$ and $\|A\|_2$ to denote the Frobenius norm and the operator norm. We denote the $\ell_\infty$-matrix norm of $A$ defined as $\nu_A = \|\!|A|\!\|_\infty \triangleq \max_{j=1,\ldots,p} \sum_{j=1}^p |A_{ij}|$. We use $\|A\|_{1,\mathrm{off}} = \sum_{i \neq j} |A_{ij}|$ to denote the off-diagonal $\ell_1$ norm. We use $\mathrm{vec}(A)$ to denote the $p^2$-vector formed by stacking the columns of $A$ and use $\Gamma(A) = (I \otimes A)$ to denote the kronecker product of $A$ with the identity matrix $I$. For symmetric positive definite matrices $A_1$ and $A_2$, we use $A_1 \succ A_2$ to denote $A_1 - A_2$ is positive definite. We define $\mathrm{sign}(A_{ij}) = +1$ if $A_{ij} > 0$ and $\mathrm{sign}(A_{ij}) = -1$ if $A_{ij} < 0$. For two-real valued functions $f(\cdot)$ and $g(\cdot)$, we write $f(n) = \mathcal{O}(g(n))$ if $f(n) \leq cg(n)$ and $f(n) = \Omega(g(n))$ if $f(n) \geq c'g(n)$ for constants $c, c' > 0$.

## 2 Problem Setup

Consider a $p$-dimensional random vector $X$ following the Gaussian distribution $\mathcal{N}(0, \Sigma_X)$ with a known covariance matrix $\Sigma_X \succ 0$ (we outline a relaxation of this assumption in Remark 1). Let $Y = (B^*)^{-1}X$ with a $p \times p$ matrix $B^* \succ 0$ and note that $Y \sim \mathcal{N}(0, \Theta^{*-1})$, where $\Theta^* = B^* \Sigma_X B^*$. Define the sample covariance matrix $S = n^{-1} \sum_{i=1}^n Y_i Y_i^\mathsf{T}$, where $\{Y_1, \ldots, Y_n\}$ are the $n$ (possibly $n < p$) i.i.d. samples of $Y$. For some $\lambda_n > 0$, we consider the $\ell_1$ regularized MLE for estimating $B^*$:

$$\underset{B \succ 0; \Theta = B\Sigma_X^{-1}B^\mathsf{T}}{\arg\min} \left[ \mathrm{Tr}(S\Theta) - \log \det(\Theta) + \lambda_n \|B\|_{1,\mathrm{off}} \right], \tag{2}$$

where $\|B\|_{1,\mathrm{off}} = \sum_{i \neq j} |B_{ij}|$ is the $\ell_1$-norm applied to the off-diagonal entries of $B \in \mathbb{R}^{p \times p}$. The loss function in (2) without the $\ell_1$ penalty is the negative log-likelihood of $Y$, and maximizing it to estimate $B^*$ yields an unrestricted MLE.

The optimization problem in (2) looks similar to the $\ell_1$-regularized log-determinant problem, which has a rich, long history in high-dimensional statistics, machine learning, signal processing, and network sciences (see for instance [35, 3, 66, 23]). The bulk of this literature focuses on estimating $\Theta^*$. The resultant estimator, referred to as the graphical LASSO (or GLASSO), has many nice theoretical properties (e.g., asymptotic consistency and support recovery in the high-dimensional regime) [48, 49, 67]. However, our estimator in (2) is significantly different from GLASSO because we are estimating $B^*$ rather than $\Theta^*$. Other studies close to our setup estimate a sparse Cholesky factor of $\Theta^*$ [27, 17, 28]. Recall that the Cholesky decomposition is given by $\Theta^* = LL^\mathsf{T}$ where $L \succ 0$ is a lower triangular matrix. We differ from this line of work on multiple fronts: (i) we do not require $B^*$ to be a lower or upper triangular Cholesky factor; (ii) our method allows for arbitrary correlations between the nodal injections resulting in an extra $\Sigma_X^{-1}$ between the factors; and (iii) to the best of our knowledge, ours is the first work to provide guarantees on the sample complexity for estimating $B^*$ in the high-dimensional regime.

**Remark 1.** *(Unknown covariance matrix $\Sigma_X$). In problem (2), we assume that $\Sigma_X$ is known. If this is not the case, we can slightly modify (2) to estimate $B^*D$ instead of $B^*$, where $D$ is the unique square root of $\Sigma_X^{-1}$ satisfying $D^2 = \Sigma_X^{-1}$. This approach works best if the sparsity of $B^*$ (approximately) equals the sparsity of $B^*D$, which for instance happens when $\Sigma_X$ is (approximately) diagonal.* □

**Remark 2.** *(On invertibility of $B^*$). The invertiblity assumption of $B^*$ ensures that $B^*$ is identifiable from samples. This holds in several applications including the ones in (2)-(4) in Section 1. However, this might not be true if $B^*$ is a Laplacian matrix that has $k$ zero eigenvalues. One common work around (see e.g., [24, 20, 21]) is to work with a reduced Laplacian matrix by deleting $k$ rows and columns of $B^*$ (we employ this insight in our experiments; see Section 4).* □

## 3 A Convex Estimator and Statistical Guarantees

In this section, we first recast the objective in (2) in terms of $B$ for a known $\Sigma_X$. We then present our main results on the performance of our estimator in (2) when $X$ is Gaussian and non-Gaussian. We comment on extending our results to other convex loss functions and conclude with an overview of the key steps in proving our results. Full details are given in the appendix.

We begin by rewriting the problem in (2) in a form that is more suitable to our methods of analysis. Let $D$ be the unique square root of $\Sigma_X^{-1}$ satisfying $D^2 = \Sigma_X^{-1}$ (see [8]). Substituting $B = B^\mathsf{T}$ and

$\Theta = BD^2B^{\mathsf{T}}$ in the cost function of (2) yields the following:

$$\widehat{B} = \underset{B \succ 0}{\arg\min} \left[ \mathrm{Tr}(DBSBD) - \log\det(B^2) + \lambda_n \|B\|_{1,\mathrm{off}} \right], \tag{3}$$

where we use the fact that the trace operator is cyclic and the determinant of a matrix product equals the product of matrix determinants. We dropped constants that have no effect on the estimate. The symmetry and invertibility of $B$ is sufficient enough to ensure that $\log(\cdot)$ is well-defined. In other words, the positive-definiteness assumption is not needed for the well-posedness of (3).

Lemma 1 below is the starting point of our analysis. It establishes two key properties of the estimator in (3) under the positive definiteness of $B$: (i) loss function in (3) is convex in $B$ and (ii) $\widehat{B}$ is unique. The following result is proved in Appendix A.3.

**Lemma** 1. (*Convexity and Uniqueness*) *For any $\lambda_n > 0$ and $B \succ 0$, if the diagonal elements of the sample covariance matrix $S_{ii} > 0$ for all $i$, then (i) the $\ell_1$-log determinant problem in (3) is convex and (ii) $\widehat{B}$ in (3) is the unique minima satisfying the sub-gradient condition $2D^2\widehat{B}S - 2\widehat{B}^{-1} + \lambda_n \widehat{Z} = 0$. Here $\widehat{Z}$ belong to the sub-gradient $\partial\|\widehat{B}\|_{1,\mathrm{off}}$ with $\widehat{Z}_{ij} = 0$, for $i = j$, and $\widehat{Z}_{ij} = \mathrm{sign}(\widehat{B}_{ij})$ when $\widehat{B}_{ij} \neq 0$ and $|\widehat{Z}_{ij}| \leq 1$ when $\widehat{B}_{ij} = 0$, for $i \neq j$.*

A few comments of Lemma 1 are in order. ( *Convexity*) First, we recall that the compositions of two convex functions is in general not convex. As an example, consider two convex functions $f(x) = x^2$ and $g(x) = -x$, however, $g(f(x)) = -x^2$ is not convex. Therefore, in light of the fact that the loss function is a composite function of $B$, it is not clear if (3) is convex. Nonetheless, in Lemma 1 we prove that (3) is a convex program. Key to our proof is the notion of monotone convex functions. (*Uniqueness*) Second, the uniqueness result is non-trivial in high-dimensions ($n < p$) because the Hessian is rank deficient, and hence, the loss function in (3) might not be be strictly convex. However, in Lemma 1 (ii) we show that $\widehat{B}$ is unique. Key to our proof is the notion of coercivity and it adapts techniques in [48] to the case where the objective function is quadratic in the optimization variable.

### 3.1 Statement of Main result

Our first result theoretically characterizes the performance of $\widehat{B}$ in (3) when $Y$ is Gaussian. Instead, our second result provides such a characterization for $\widehat{B}$ when $Y$ is non-Gaussian. At a crude level, our results guarantee that when the number of samples $n$ scales as $d^2 \log p$, our $\ell_1$-regularized MLE correctly recovers the support of $B^*$ and is close to $B^*$ (measured in Frobenius and operator norms) with high probability. Here, $d$ is the degree of the graph underlying $B^*$.

Since we consider an $\ell_1$ regularized log-determinant program for our ML estimator (3), our results might appear similar to that of [48]. However, as also pointed in Section 3.2, our main results, including the assumptions and sufficient conditions needed to derive them, are not subsumed by those in [48], or vice versa (see Remark 3 and Section 4 for more thorough discussion).

We begin with the a few assumptions that are essential to prove our theoretical statements. Similar subset of assumptions in the context of $\ell_1$ regularized least squares problem appeared in [63, 57, 39, 68], and in the context of $\ell_1$ regularized inverse covariance estimation problem appeared in [48, 67]. We define the edge set $\mathcal{E}(B^*) = \{(i, j) : B^*_{ij} \neq 0, \text{for all } i \neq j\}$. Let $E := \{\mathcal{E}(B^*) \cup (1,1) \ldots \cup (p,p)\}$ be the augmented set including the diagonal elements. Let $E^c$ be the complement of $E$.

**[A1] Mutual incoherence condition.** Let $\Gamma^*$ be the Hessian of the log-determinant function in (3):

$$\Gamma^* \triangleq \nabla_B^2 \log\det(B)|_{B=B^*} = B^{*-1} \otimes B^{*-1}. \tag{4}$$

For $\Gamma^*$ in (4), there exists some $\alpha \in (0, 1]$ such that, $\left\|\left\|\Gamma^*_{E^c E}(\Gamma^*_{EE})^{-1}\right\|\right\|_\infty \leq 1 - \alpha$.

**[A2] Hessian regularity condition.** Let $d$ be the maximum number of non zero entries among all the rows in $B^*$ (i.e., the degree of the underlying graph), $\Theta^* = B^*\Sigma_X^{-1}B^*$, and $D^2 = \Sigma_X^{-1}$. Then,

$$\left\|\left\|\Gamma^{*-1}\right\|\right\|_\infty \leq \frac{1}{4d\|\Theta^{*-1}\|_\infty \|D^2\|_\infty}. \tag{5}$$

A few comments are in order. **[A1]** Our novel mutual incoherence condition on $B^*$ regulates the influence of irrelevant variables (elements of Hessian restricted to $E^c \times E$) on relevant variables

(elements of Hessian restricted to $E \times E$). The $\alpha$-incoherence assumption of the above type is standard in literature, and [48] demonstrates its validity for several graphs, including chain and grid graphs, which we will explore in experimental section. Notice that the $\alpha$-incoherence in [48] is imposed on $\Theta^*$. Instead, we require it on $B^*$. [A2] This condition is in parallel with bounding the maximum eigenvalue of $\Theta^{*-1}$ condition for estimating sparse $\Theta^*$ (see for e.g., [49, 30]).

Our problem set-up assumes that the injected flows $X_i$, $i \in [p]$, are Gaussian. Given that the vector of node potentials $Y$ depends on $X$ via the *balance equation* (see equation (1)), we have that $Y$ is Gaussian. However, we work with sub-Gaussian distributions, a natural generalization to the Gaussian case, which encompasses many well known distributions that occur in practice (for e.g., bounded random variables, gaussians and mixture of gaussians). We define this distributional assumption below.

**Definition 1.** *(Sub-Gaussian random variable) A zero mean random variable $Z$ is said to be sub-Gaussian if there exists a constant $\sigma > 0$ such that for any $t \in \mathbb{R}$, $\mathbb{E}[\exp(tZ)] \leq \exp\left(\sigma^2 t^2/2\right)$.*

Our first main result below provides sufficient conditions on the number of samples $n$ needed for $\widehat{B}$ in (3) to exactly recover the sparsity structure of $B^*$ and to achieve sign consistency, defined as $\text{sign}(\widehat{B}_{ij}) = \text{sign}(B^*_{ij})$, for all $(i,j) \in E$. We recall that $\nu_A = \|A\|_\infty \triangleq \max_{j=1,\dots,p} \sum_{j=1}^p |A_{ij}|$ and define $\Sigma^* = \Theta^{*-1}$ to be the covariance matrix of the node potential $Y$.

**Theorem 1.** *(Support Recovery: Sub-Gaussian) Let $X = (X_1, \dots, X_p)$ be the vector of injected flows. Suppose that for all $i$, $X_i/\sqrt{\Sigma_X(ii)}$ is sub-Gaussian with parameter $\sigma$ and assumptions [A1-A2] hold. Let the regularization parameter $\lambda_n = C_0\sqrt{\tau(\log 4p)/n}$, where $C_0$ is given below. Given $n$ independent samples from $Y$, if the sample size $n > C_1^2 d^2(\tau \log p + \log 4)$, the following hold with probability at least $1 - \frac{1}{p^{\tau-2}}$, for some $\tau > 2$:*

(a) $\widehat{B}$ *exactly recovers the sparsity structure of* $B^*$*; that is,* $\widehat{B}_{E^c} = 0$,

(b) $\widehat{B}$ *satisfies the element-wise* $\ell_\infty$ *bound* $\|\widehat{B} - B^*\|_\infty \leq C_2\sqrt{\frac{\tau \log p + \log 4}{n}}$*, and*

(c) $\widehat{B}$ *satisfies sign consistency if* $|B^*_{\min}| \geq 2C_2\sqrt{\frac{\tau \log p + 4}{n}}$*,* $B^*_{\min} \triangleq \min_{(i,j) \in \mathcal{E}(B^*)} |B^*_{ij}|$,

*where* $C_1 = 192\sqrt{2}[(1+4\sigma^2)\kappa(\Sigma^*)\text{Tr}(\Sigma_X)\nu_{D^2}\nu_{B^*}\nu_{B^{*-1}}]\max\{\nu_{\Gamma^{*-1}}\nu_{B^{*-1}}, 2\nu_{\Gamma^{*-1}}^2\nu_{B^{*-1}}^3, 2\alpha^{-1}d^{-1}\}$, $C_2 = [64\sqrt{2}(1 + 4\sigma^2)\kappa(\Sigma^*)\text{Tr}(\Sigma_X)\nu_{\Gamma^{*-1}}\nu_{D^2}\nu_{B^*}\nu_{B^{*-1}}]$, $C_0 = C_2/(4\nu_{\Gamma^{*-1}})$ *and* $\kappa(\cdot)$ *is the condition number.*

The quantities $(\nu_{\Gamma^{*-1}}, \nu_{D^2}, \nu_{B^*}, \kappa(\Sigma^*), \text{Tr}(\Sigma_X))$ capture the inherent complexity of the model and do not depend on the number of samples $n$. As long as the magnitude of the entries in $\Gamma^{*-1}, D^2$, and $B^*$ scale as $\mathcal{O}(1/d)$, the model complexity parameters do not depend on $(p, d)$. That is, as the size of the network grows with $(p, d)$ the edge strengths decay with $d$. Suppose that the model complexity parameters are constants and that $n = \Omega(d^2 \log p)$. Then part (a) of Theorem 1 guarantees that our ML estimator does not falsely include entries (or edges in the underlying graph) that are not in the support of $B^*$. Part (b) establishes the element-wise $\ell_\infty$ norm consistency of $\widehat{B}$; that is, $\|\widehat{B} - B^*\|_\infty = \mathcal{O}(\sqrt{(\log p)/n})$. Finally, part (c) establishes sign consistency of $\widehat{B}$, and hence, our estimator does not falsely exclude entries that are in the support of $B^*$. Crucial is the requirement of $|B^*_{\min}| = \Omega\left(\sqrt{(\log p)/n}\right)$, which puts a limit on the minimum (in absolute) value of the entries in $B^*$. This condition parallels the familiar *beta-min* condition in the LASSO literature (see [63, 59]).

We now present a corollary to Theorem 1 that gives consistency rates of convergence for $\widehat{B}$ in the Frobenius and operator norms. Let $\mathcal{E}(B^*) = \{(i,j) : B^*_{ij} \neq 0, \text{for all } i \neq j\}$ be the edge set of $B^*$.

**Corollary 1.** *Let $s = |\mathcal{E}(B^*)|$ be the cardinality of $\mathcal{E}(B^*)$. Under the same hypotheses in Theorem 1, with probability greater than $1 - \frac{1}{p^{\tau-2}}$, the estimator $\widehat{B}$ satisfies*

$$\|\widehat{B} - B^*\|_F \leq C_2\sqrt{\frac{(s+p)(\tau \log p + 4)}{n}} \quad \text{and} \quad \|\widehat{B} - B^*\|_2 \leq C_2 \min\{d, \sqrt{s+p}\}\sqrt{\frac{\tau \log p + 4}{n}}.$$

*Proof sketch.* Both the Frobenius and operator norm bounds follows by applying standard matrix norm inequalities to the $\ell_\infty$ consistency bound in part (b) of Theorem 1. Importantly, $s + p$ is the

bound on maximum number of non-zero entries in $B^*$, where $s$, by definition, is the total number of off-diagonal non-zeros in $B^*$. Complete details are provided in Appendix A.3.

Thus far we have assumed that the nodal potentials $Y_i$ are sub-Gaussian random variables. We now explore another broad class of random variables with bounded $k^{\text{th}}$ moments, which are known to have tails that decay according to some power law [44]. An important example of power law distributions are Pareto distributions which finds applications in a wide variety of areas [43, 41]. Motivated by such important practical considerations, we state our next result for random variables with bounded moments. We begin with the following definition.

**Definition 2.** *(Bounded moments) A random variable $Z$ is said to have bounded $4k^{th}$ moment if there exists a constant $M_k \in \mathbb{R}$ such that $\mathbb{E}\left[(Z)^{4k}\right] \leq M_k$.*

Results below parallel Theorem 1 and Corollary 1 for random variables with bounded moments.

**Theorem 2.** *(Support Recovery: Bounded Moments) Let $X = (X_1, \ldots, X_p)$ be the vector of injected flows such that for all $i$, the node potential $Y_i/\sqrt{\Sigma_{ii}^*}$ has bounded moment as in Definition 2 and assumptions [A1-A2] hold. Let the regularization parameter $\lambda_n = C_0\sqrt{\tau(\log 4p)/n}$, with $C_0$ defined in Theorem 1. Given $n$ independent samples from $Y$, if the sample size $n > C_4 d^2 p^{\tau/k}$, then with probability greater than $1 - 1/p^{\tau-2}$, for some $\tau > 2$, the following hold: (a) $\widehat{B}$ exactly recovers the sparsity structure of $B^*$ (that is $\widehat{B}_{E^c} = 0$); (b) the element-wise $\ell_\infty$ bound $\|\widehat{B} - B^*\|_\infty \leq C_5\sqrt{\frac{p^{\tau/k}}{n}}$; and (c) $\widehat{B}$ satisfies sign consistency if $|B_{\min}^*| \geq 2C_5\sqrt{\frac{p^{\tau/k}}{n}}$.*

The constants and their dependence on the model complexity parameters are given in Appendix A.3.

**Corollary 2.** *Suppose the hypotheses in Theorem 2 hold. Then with probability greater than $1 - \frac{1}{p^{\tau-2}}$:*
$$\|\widehat{B} - B^*\|_F \leq C_5\sqrt{\frac{(s+p)(p^{\tau/k})}{n}} \text{ and } \|\widehat{B} - B^*\|_2 \leq C_5 \min\{d, \sqrt{s+p}\}\sqrt{\frac{p^{\tau/k}}{n}}, \text{ where } s = |\mathcal{E}(B^*)|.$$

Interpretations of Theorem 1 and Corollary 1 also hold for Theorem 2 and Corollary 2. However, in this setting, we have different sample size $n = \Omega(d^2 p^{\tau/k})$ and $|B_{\min}^*| = \Omega(\sqrt{p^{\tau/k}/n})$, where $k$ is given by Definition 2. In contrast, for sub-Gaussian case we have logarithmic dependence in $p$ (the number of vertices). Finally, albeit fundamentally different from GLASSO estimator, we were able to obtain consistency rates for $\widehat{B}$ (3) that are similar to those in [48, 12].

**Remark 3.** *(Comparison with the GLASSO estimator). For simplicity, suppose that $\Sigma_X$ is diagonal. Then, it follows that $B^* \succ 0$ is the unique square root of $\Theta^* = (B^*)^2$. Thus, a naïve way to estimate $B^*$ is by taking the square root of the GLASSO estimate $\widehat{\Theta}$. Let us call this estimator $\widehat{B}_{SR}$ and note that $\widehat{B}_{SR}$ inherits its optimal properties from $\widehat{\Theta}$. We show that $\widehat{\Theta}$ has sub-optimal estimation rate than $\widehat{B}$ in (3) for estimating $B^*$. Let $B^*$ contains $d$ non-zero elements in every row. Then the underlying graph of $\Theta^*$ is a two-hop network with degree $d^2$. Using sample complexity results from [48], it follows that $\widehat{\Theta}$ requires $n = \Omega(d^4 \log p)$ to estimate $B^*$. Instead, our $\ell_1$-regularized MLE requires $n = \Omega(d^2 \log p)$ samples. This reduction is more pronounced for networks with a large degree $d$.* □

### 3.2 Outline of Main Analysis

We provide an outline of our methods and main strategies to prove Theorem 1. We employ the *primal-dual witness technique*—a well-known method used derive to statistical guarantees for sparse convex estimators [63, 64, 34]. This technique involves constructing a primal-dual pair $(\widetilde{B}, \widetilde{Z})$ satisfying the zero-subgradient condition of the convex problem in (3), such that (the primal) $\widetilde{B}$ has the correct (signed) support. Suppose this construction succeeds, from the uniqueness result in Lemma 1, it follows that $\widehat{B} = \widetilde{B}$, and the dual $\widetilde{Z}$ is an optimal solution to the dual of (1). Thus, at the heart of our analysis is in showing that the primal-dual construction succeeds with high-probability. Similar technique is also used to prove Theorem 2 (i.e., the non-Gaussian case); see Appendix A.3.

While our proof methods are inspired from [48, 63], our analysis is more involved due to the presence of $B^2$, as opposed to $B$, in the loss function of (3). Consequently, we require more nuanced assumption (as in [A2]) and dual feasiblity condition than the ones in [48] (see below).

### 3.3 Primal-dual pair and supporting lemmas

We briefly introduce the primal-dual witness construction. In Lemma 2, we provide sufficient conditions under which this construction succeeds.

We construct the primal-dual pair $(\widetilde{B}, \widetilde{Z})$ as follows. The primal solution $\widetilde{B}$ is determined by solving

$$\widetilde{B} \triangleq \underset{B=B^T, B \succ 0, B_{E^c}=0}{\arg\min} \left[ \text{Tr}(DBSBD) - \log\det(B^2) + \lambda_n \|B\|_{1,\text{off}} \right]. \tag{6}$$

Here (6) is a restricted problem, in that, we impose $B_{E^c} = 0$. Also, we have $\widetilde{B} \succ 0$ and $\widetilde{B}_{E^c} = 0$. The dual $\widetilde{Z} \in \partial\|\widetilde{B}\|_{1,\text{off}}$ is chosen such that it satisfies the zero-subgradient condition of (6). This is obtained by setting $2\lambda_n \widetilde{Z}_{ij} = [\widetilde{B}^{-1}]_{ij} - [D^2\widetilde{B}S]_{ij}$, for all $(i,j) \in E^c$. It can be verified that $(\widetilde{B}, \widetilde{Z})$ satisfies the zero-subgradient condition (see the statement of Lemma 1) of the original problem in (3). Thus, it remains to establish the strict dual feasibility condition; that is $|\widetilde{Z}_{ij}| < 1$, for any $(i,j) \in E^c$.

We introduce some notation. Let $W \triangleq S - \Theta^{*-1}$ be a measure of noise in the data, where $S$ is the sample covariance and $\Theta^{*-1}$ is the true covariance of $Y$. Let $\Delta \triangleq \widetilde{B} - B^*$ be a measure of distortion between the primal solution $\widetilde{B}$ as defined in equation (6) and the true matrix to be estimated $B^*$. We also need the the higher order terms (denoted by $R(\Delta)$) of the Taylor expansion of the gradient $\nabla \log\det(\widetilde{B})$ centered around $B^*$ [10]:

$$\nabla \log\det(\widetilde{B}) = B^{*-1} + B^{*-1}\Delta B^{*-1} + \underbrace{\widetilde{B}^{-1} - B^{*-1} - B^{*-1}\Delta B^{*-1}}_{\triangleq R(\Delta)}. \tag{7}$$

**Lemma 2.** *(Sufficient conditions for strict dual feasibility) Let the regularization parameter $\lambda_n > 0$ and $\alpha$ be defined as in* **[A1]**. *Suppose the following holds*

$$\max\left\{ \||\Gamma(D^2\Delta) + \Gamma(D^2B^*)\||_\infty \|W\|_\infty, \|R(\Delta)\|_\infty, \||\Gamma(D^2\Delta)\||_\infty \|\Theta^{*-1}\|_\infty \right\} \leq \frac{\lambda_n\alpha}{24}. \tag{8}$$

*Then the dual vector $\widetilde{Z}_{E^c}$ satisfies $\|\widetilde{Z}_{E^c}\|_\infty < 1$, and hence, $\widetilde{B} = \widehat{B}$.*

*Proof sketch*: The proof essentially involves expressing the sub-gradient condition in Lemma 1 as a vectorized form using $R(\Delta)$ (in (7)) and $W$. By manipulating the vectorized sub-gradient condition, we obtain an expression of $\widetilde{Z}_{E^c}$ that is a function of the quantities in (8). We finish off the proof by repeated applications of triangle inequality of norms and invoking assumptions in Lemma (8).

The following results provides us with dimension and model complexity dependent bounds on the reminder term $R(\Delta)$ in (7) and the distortion $\Delta$.

**Lemma 3.** *(Control of reminder) Suppose that the element-wise $\ell_\infty$-bound $\|\Delta\|_\infty \leq \frac{1}{3\nu_{B^{*-1}}d}$ holds, where $\nu_{B^{*-1}} = \||B^{*-1}\||_\infty$. Then $\|R(\Delta)\|_\infty \leq \frac{3}{2}d\|\Delta\|_\infty^2 \nu_{B^{*-1}}^3$.*

The proof, adapted from [48], is algebraic in nature and relies on certain matrix expansions. The details are provided in Appendix. In the following result, we provide a sufficient condition under which the element-wise $\ell_\infty$-bound on $\Delta$ in Lemma 3 holds.

**Lemma 4.** *(Control of $\Delta$) Let $r \triangleq 4\nu_{\Gamma^{*-1}} [\nu_{D^2}\nu_{B^*}\|W\|_\infty + 0.5\lambda_n] \leq \min\left\{ \frac{1}{3\nu_{B^{*-1}}d}, \frac{1}{6\nu_{\Gamma^{*-1}}\nu_{B^{*-1}}^3 d} \right\}$. Then we have the element-wise $\ell_\infty$ bound $\|\Delta\|_\infty = \|\widetilde{B} - B^*\|_\infty \leq r$.*

*Proof sketch*: By construction $\widetilde{B}_{E^c} = B^*_{E^c} = 0$. Hence, $\|\Delta\|_\infty = \|\Delta_E\|_\infty$, where $\Delta_E = \widetilde{B}_E - B^*_E$. We construct a continous vector valued function $F : \Delta_E \to \Delta_E$ that has a unique fixed point. Invoking assumptions **[A1]-[A2]**, we show that $F(\cdot)$ is a contractive map on the $\ell_\infty$ ball defined as $\mathbb{B}_r = \{A : \|A\|_\infty \leq r\}$ with $r$ defined in the statement of the lemma. Specifically, we show that $F(\mathbb{B}_r) \subseteq \mathbb{B}_r$. Finally, we finish off the proof by an application of Brower's fixed point theorem [29] to show that the unique fixed point is inside $\mathbb{B}_r$. Consequently, $\|\Delta\|_\infty \leq r$.

Finally, the result in Theorem 1 follows by putting these lemmas together for an appropriate choice of $\lambda_n$ and the sample size requirement, and there upon, invoking some known concentration inequalities. Refer Appendix A.3 for complete details.

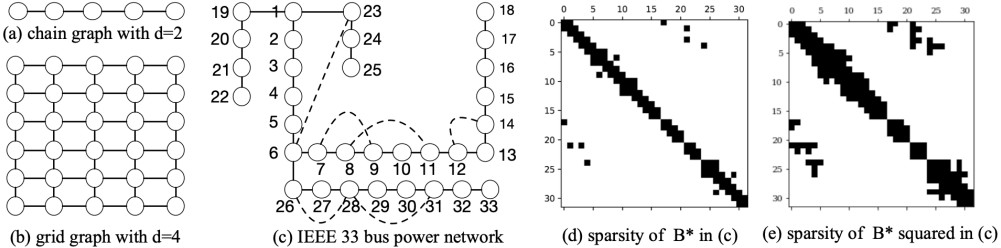

Figure 2: Graphs used in experiments. (a) Chain graph with maximum degree $d = 2$. (b) Grid graph $d = 4$. (c) IEEE 33 bus (node) distribution network with additional loops (shown in dashed lines). (d) Sparsity of $B^*$ associated with the IEEE 33 bus network. (d) Sparsity of $(B^*)^2$. Notice that $(B^*)^2$ is denser relative to $B^*$. Consequently, GLASSO+HR needs more samples than $\ell_1$-MLE to recover the support (see plots in Fig. 3).

## 4  Experiments

We validate the support recovery performance of our $\ell_1$-regularized MLE on synthetic and a benchmark power distribution network (see Fig. 2). We choose $\lambda_n$ proportional to $\sqrt{\log p/n}$. Our results are averaged over 100 trials of $n$ independent samples of $Y$. We compare $\ell_1$-regularized MLE performance with (i) the square-root estimator (hereafter, GLASSO+SR) that identifies the support of $B^*$ by determining $(i, j)$ for which $\widehat{\Theta}_{i,j}^{\frac{1}{2}} \neq 0$; and (ii) the GLASSO+2HR (Hop Refinement) estimator [20] that identifies the support of $B^*$ by determining $(i, j)$ for which $\widehat{\Theta}_{i,j} \leq -\tau$ for $\tau = 1e-02$. Here $\widehat{\Theta}$ is the GLASSO estimate of the inverse covariance matrix of $Y$ [23]. These estimators are described in detail in Introduction.To have a fair comparison with the GLASSO based estimators, we set $\Sigma_X$ ($X$ is the injected vector) to be diagonal. However, as discussed earlier, our $\ell_1$ regularized ML estimator works for any $\Sigma_X \succ 0$. We consider $p$ to be as large as 64 nodes, Computational examples involving large data matrices for $B^*$ having a lower triangular matrix form has been reported in [28].

*(i) Synthetic data*: We consider two undirected graphs for $B^*$, the chain graph and the grid graph for $p = \{32, 64\}$ nodes. We set $B_{ij}^* = 1$ for $(i, j) \in E$ and $B_{ij}^* = 0$ for $(i, j) \in E^c$, where $E$ can be the edge set of the chain or grid graph. We then adjust the diagonal elements of $B^*$ to ensure $B^* \succ 0$.

*(ii) Power network:* We set $B^*$ to be the Laplacian of the IEEE 33 bus power distribution network [69]. For this data, we note that $B^*$ is non-invertible because of one zero eigenvalue. We obtain the reduced $B^*$ by deleting the first row and column of $B^*$. We also slightly modify the network by adding three loops of cycle length three, two of cycle length four, and one loop of cycle length five (see Fig 2). We made these modifications to highlight that $\ell_1$-MLE imposes no connectivity assumptions on the graph underlying $B^*$, except sparsity. In contrast, GLASSO+2HR estimator [20] restricts the graph underlying $B^*$ from having cycles of length three (i.e., triangle-free).

In Fig. 3, we show empirical support recovery probabilities for all three estimators as a function of the number of samples $n$. Both on synthetic and power network data, $\ell_1$-MLE achieved superior rates than the other competing estimators. In fact, $\ell_1$-MLE exactly recovers the support of $B^*$ when the number of samples is in the order of $d^2 \log p$, which is in excellent agreement with the proposed theory. Instead, for a similar performance, GLASSO+SR needed $d^4 \log p$ samples (see Remark 3).

## 5  Discussions and Future Work

High dimensional networks obeying conservation laws of the form $X = B^*Y$ are often used to model and study interactions among different conserved quantities in various engineering and scientific disciplines. For such systems, with unknown structure of the network, or equivalently the sparsity pattern of $B^*$, we estimate the the sparsity structure of $B^*$ using an $\ell_1$-regularized ML estimator. Our estimator relies on $n$ samples of the node potentials $Y$, with some knowledge of the statistics of the node injections $X$. We showed that this estimator is the unique optimal solution to a variant of the convex log-determinant program. Using novel mutual incoherence conditions, we have provided (theoretical) sparsity and support recovery consistency for our estimator. Finally, using several numerical results we not only validated our theory but also showed that our $\ell_1$-regularized estimator outperforms methods that employ GLASSO based estimators.

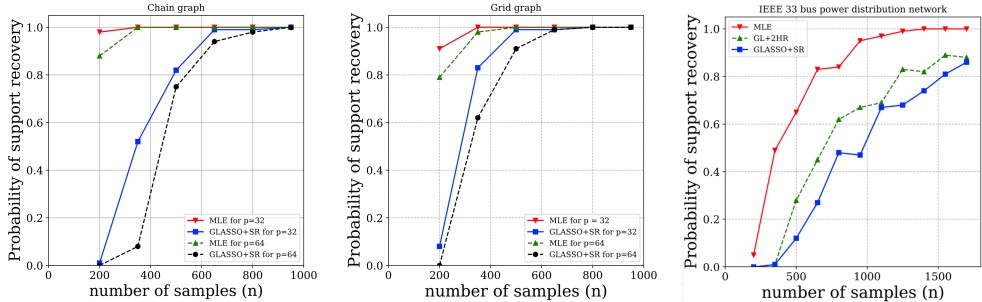

Figure 3: Empirical probability of success of various estimators versus the raw sample size $n$ for (left) chain graph, (middle) grid graph, and (right) IEEE 33 bus network. For chain and grid graph, we compare our $\ell_1$ regularized MLE performance with GLASSO+SR for $p \in \{32, 64\}$. Instead, for IEEE 33 bus network, we compare $\ell_1$ regularized MLE with GLASSO+SR and GLASSO+2HR.

In our setup, we neither require actual injected flows $(X)$, nor $B^*$ to be a Laplacian matrix, nor $n > p$, thereby allowing our learning problem to be general enough to be applicable for a variety of domains ranging from electrical networks to social networks. Consequently, our framework and theoretical results admits many future extensions and refinements. For instance, verifying the assumptions [**A1-A2**] in practice is a fruitful future direction. It is worth noting that, in order to run the proposed algorithm/estimator, one need not verify the assumptions. Notice that from Lemma 1, the loss function/objective is convex and admits a unique solution if the regularization constant $\lambda_n > 0$ and $B$ is positive definite, therefore the unknown structure of the network can be recovered by finding the unique minima. However, in order to theoretically guarantee the properties of the estimator mentioned in Theorem 1 and Theorem 2, assumptions [**A1-A2**] are needed. A possible extension to the current problem is to recast the objective function in (2) as the minimization of the Bregman divergence for more general loss functions. Another promising direction is to generalize (2) to include non-invertible network matrix $B^*$. We leave this as our future work.

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
