# A Appendix

## A.1 Code Availability

In Section 4, we performed numerical experiments to validate the support recovery property of the proposed $\ell_1$ regularized ML estimator and compare its performance with the square root estimator (GLASSO+SR) and what we call the GLASSO+2HR (2 Hop Refinement) estimator [20]. Our benchmark examples include a synthetic data for chain and grid graphs, and a real-world electric power system network. We implement all three estimators using CVXPY 1.2 open source python package on Google Colab. All the simulation results reported in this paper can be reproduced using the code available at https://github.com/AnirudhRayas/SLNSCL.

## A.2 Limitations

In this work we restricted $B^*$ in the model $B^*Y - X = 0$ to be invertible and positive definite. In applications such as transportation networks, $B^*$ might not be symmetric, and hence, not positive definite. For this setting, it is not clear if our technical analysis hold. Another limitation is that the our sample complexity result depends quadratically in the maximum degree $d$. This is because of the proof technique we adapted from [48]. Decreasing this dependence from $d^2$ to $d$ is an open question. Finally, as is well known, verifying regularity conditions, such as the mutual incoherence,[2] in practice is computationally hard. Hence, there is a need to develop regularity conditions that are easily verifiable in practice.

## A.3 Proofs of all technical results

**Overview**: We begin with a brief overview of the problem set-up and state the necessary assumptions. Then, we provide proofs for all the technical results. Recall that our observation model is $Y = B^{*-1}X$, where $B^*$ is a $p \times p$ sparse matrix which encodes the structure of a network with the property that $B^*_{ij} = 0$ for all $(i,j) \in E^c$, $Y \in \mathbb{R}^p$ is the vector of node potentials and $X \in \mathbb{R}^p$ is the unknown random vector of injected flows with known covariance matrix $\Sigma_X$. Given $n$ i.i.d samples of the vector $Y$ our goal is to learn the sparsity structure of the matrix $B^*$. Towards this we propose an estimator $\widehat{B}$ which is the solution of the following $\ell_1$ regularized log-det problem

$$\widehat{B} = \underset{B \succ 0}{\arg\min} \left[ \text{Tr}(DBSBD) - \log\det(B^2) + \lambda_n \|B\|_{1,\text{off}} \right]. \tag{9}$$

where $D \in \mathbb{R}^{p \times p}$ is the unique square root of $\Sigma_X^{-1}$ and $S$ is the sample covariance matrix constructed from $n$ samples of the random vector $Y$. We recall the assumptions necessary to prove our results.

**[A1] Mutual incoherence condition.** Let $\Gamma^*$ be the Hessian of the log-determinant function in (9):

$$\Gamma^* \triangleq \nabla_B^2 \log\det(B)|_{B=B^*} = B^{*-1} \otimes B^{*-1}. \tag{10}$$

For $\Gamma^*$ in (10), there exists some $\alpha \in (0,1]$ such that $\left\| \Gamma^*_{E^c E}(\Gamma^*_{EE})^{-1} \right\|_\infty \leq 1 - \alpha$.

**[A2] Hessian regularity condition.** Let $d$ be the maximum number of non zero entries among all the rows in $B^*$ (i.e., the degree of the underlying graph), $\Theta^* = B^* \Sigma_X^{-1} B^*$, and $D^2 = \Sigma_X^{-1}$. Then,

$$\left\| \Gamma^{*-1} \right\|_\infty \leq \frac{1}{4d\|\Theta^{*-1}\|_\infty \|D^2\|_\infty}. \tag{11}$$

Our analysis is based on the Primal-Dual Witness (PDW) construction to certify the behaviour of the estimator $\widehat{B}$. The PDW technique consists of constructing a primal-dual pair $(\widetilde{B}, \widetilde{Z})$, where $\widetilde{B}$ is the primal solution of the restricted log-det problem defined below

$$\widetilde{B} \triangleq \underset{B=B^T, B \succ 0, B_{E^c}=0}{\arg\min} \left[ \text{Tr}(DBSBD) - \log\det(B^2) + \lambda_n \|B\|_{1,\text{off}} \right]. \tag{12}$$

where $\widetilde{Z}$ is the optimal dual solution. By definition the primal solution $\widetilde{B}$ satisfies $\widetilde{B}_{E^c} = B^*_{E^c} = 0$. Furthermore the pair $(\widetilde{B}, \widetilde{Z})$ are solutions to the zero gradient conditions of the restricted problem

---

[2] Interestingly, this condition is necessary and sufficient for sparse linear regression problems [63].

([12](#)). Therefore, when the PDW construction succeeds the solution $\widehat{B}$ is equal to the primal solution $\widetilde{B}$ which guarantees the support recovery property ie. $\widehat{B}_{E^c} = 0$.

We now summarize our technical results. ❶ We begin by showing that the $\ell_1$ regularized log-det problem in ([9](#)) is convex and admits a unique solution $\widehat{B}$ (see Lemma [1](#)). ❷ We then proceed to derive the sufficient conditions under which the PDW construction succeeds (see Lemma [2](#)). ❸ We then guarantee that the remainder term $R(\Delta)$ is bounded if $\Delta$ is bounded (see Lemma [3](#)). ❹ Furthermore, for a specific choice of radius $r$ as a function of $\|W\|_\infty$ we show that $\Delta$ lies in a ball $\mathbb{B}_r$ of radius $r$ (see Lemma [4](#)). ❺ We then derive a lemma which we call *the master lemma* which gives support recovery guarantees and element-wise $\ell_\infty$ norm consistency for our estimator $\widehat{B}$ under no specific distributional assumptions (see Lemma [A.6](#)). ❻ Using known concentration results on sub-gaussian and moment bounded random vectors we prove our main result for the two distributions mentioned above. Recall that our main result gives sufficient conditions on the number of samples required for our estimator $\widehat{B}$ to recover the exact sparsity structure of $B^*$. We also show that under these sufficient conditions $\widehat{B}$ is consistent with $B^*$ in the element-wise $\ell_\infty$ norm and achieves sign consistency if $|B^*_{\min}|$ (the minimum non-zero entries of $B^*$) is lower bounded (see Theorem [1](#) and Theorem [2](#)). ❼ Finally, we show that $\widehat{B}$ is consistent in the Frobenius and spectral norm.

**Numbering convention**: To make the appendix self contained we restated statements of all theorems, lemmas, and definitions with their numbers unchanged with respect to the main text. For the numbered environments that are specifically introduced in Appendix, the environment begins with the label "A" (e.g., Lemma A.1).

**Lemma 1.** (*Convexity and uniqueness*) *For any $\lambda_n > 0$ and $B \succ 0$, if the diagonal elements of the sample covariance matrix $S_{ii} > 0$ for all $i$, then (i) the $\ell_1$-log determinant problem in ([9](#)) is convex and (ii) $\widehat{B}$ in ([9](#)) is the unique minima satisfying the sub-gradient condition $2D^2\widehat{B}S - 2\widehat{B}^{-1} + \lambda_n\widehat{Z} = 0$. Here $\widehat{Z}$ belong to the sub-gradient $\partial\|\widehat{B}\|_{1,\text{off}}$ so that $\widehat{Z}_{ij} = 0$, for $i = j$, and $\widehat{Z}_{ij} = \text{sign}(\widehat{B}_{ij})$ when $\widehat{B}_{ij} \neq 0$ and $|\widehat{Z}_{ij}| \leq 1$ when $\widehat{B}_{ij} = 0$, for $i \neq j$.*

*Proof.* *(i) Convexity*: Let $S = MM^T$, for some $M \succeq 0$, and recall that $\|A\|_F^2 = \text{Tr}(AA^\mathsf{T})$. Then, the objective function in ([9](#)) can be expressed as

$$\|DBM\|_F^2 - \log\det(B^2) + \lambda_n\|B\|_{1,\text{off}}. \tag{13}$$

First, the square-root of the first term is convex because for any $\lambda \in (0,1)$ and $B_1, B_2 \succ 0$, we have

$$\|D(\lambda B_1 + (1-\lambda)B_2)M\|_F = \|\lambda DB_1M + (1-\lambda)DB_2M\|_F$$
$$\leq \lambda\|DB_1M\|_F + (1-\lambda)\|DB_1M\|_F.$$

Now that $h_1(x) = x^2$ and $h_2(A) = \|A\|_F$ are both convex and that $h_1(x)$ is non-decreasing on the range of $h_2$, that is, $[0,\infty]$, it follows that the composition $h_1 \circ h_2 = \|\cdot\|_F^2$ is convex.

Second, we show the convexity of $-\log\det(B^2)$ using the perspective function technique [[10](#)]. To this end, let $|\cdot|$ be the absolute value and note that $\log\det(B^2) = \log|\det(B^2)| = 2\log|\det(B)|$. Let $g(t) = \log|\det(B + tV)|$ with $V \succeq 0$ be the perspective function of $\log|\det(B)|$. Since $B$ is symmetric and invertible, there exists an orthogonal matrix $Q$ such that $QQ^\mathsf{T} = I$ and $B = Q\Omega Q^\mathsf{T}$, where $\Omega$ is a diagonal matrix consisting of eigenvalues of $B$. Then,

$$g(t) = \log(|\det(Q\Omega Q^\mathsf{T} + tQQ^\mathsf{T}VQQ^\mathsf{T})|) \tag{14}$$
$$= \log(|\det(Q(\Omega + tQ^\mathsf{T}VQ)Q^\mathsf{T})|) \tag{15}$$
$$= \log(|\det(\Omega + tQ^\mathsf{T}VQ)|) \tag{16}$$
$$= \log(|\det(I + t\Omega^{-1}Q^\mathsf{T}VQ)|) + \log(|\Omega|), \tag{17}$$

where we used the facts $|\det(X_1X_2)| = |\det(X_1)\det(X_2)| = |\det(X_1)||\det(X_2)|$ and $\Lambda$ is full rank. Since $\Omega$ is diagonal and $Q^\mathsf{T}VQ \succeq 0$, it follows that the eigenvalues $\{\lambda_i\}$ of $\Omega^{-1}Q^\mathsf{T}VQ$ are real-valued (need not be positive). Thus,

$$g(t) = \log\prod|(1 + t\lambda_i)| + \log(|\Omega|).$$

Notice that $g'(t) = \sum \frac{\lambda_i(1+t\lambda_i)}{(1+t\lambda_i)^2}$ and $g''(t) = -\sum \frac{\lambda_i^2}{(1+t\lambda_i)^2} < 0$. Thus $g(t)$ is strictly concave. Hence, $2\log|\det(B)| = \log\det(B^2)$ is strictly concave. Finally, $-\log\det(B^2)$ is strictly convex.

Third, the norm $\lambda_n\|B\|_{1,\text{off}}$ is the sum of absolute values of off-diagonal terms, and hence, convex. Because the sum of convex functions and a strictly convex function is strictly convex, we conclude that the objective function in (14) is strictly convex.

**Remark**: In the proof, we used the fact that $B$ is symmetric and full rank but not the positive-definite. The proof for $B \succ 0$ is simple because we can drop the absolute values and mimic the standard log-det concavity proof [48, 23]. Finally, we required $V \succeq 0$ to not to deal with the (possible) imaginary eigenvalues of $\Omega^{-1}Q^{\mathsf{T}}VQ$. However, we conjecture that $V$ needs to be only symmetric.

*(ii) Uniqueness*: In part (i), we showed that the objective function in (14) is strictly convex. Recall that strictly convex functions have the property that the minimum is unique if attained [10]. We show that the minimum is attained using the notion of coercivity (see Def 11.10 and Proposition 11.14 in [5]). This amounts to showing that the objective function $CO(B) \triangleq (\|DBM\|_F^2 - 2\log\det B)$ subject to constraints (see below) tend to infinity as $\|B\|_2 \to \infty$.

By Lagrangian duality, the $\ell_1$ regularized log-det problem (9) can be written as

$$\underset{B\succ 0, B=B^T, \|B\|_{1,\text{off}}<\Lambda_n}{\arg\min} \|DBM\|_F^2 - 2\log\det B, \tag{18}$$

where $\Lambda_n$ is the constraint on the off diagonal elements of $B$. From the constraint $\|B\|_{1,\text{off}} < \Lambda_n$, it follows that the off-diagonal elements of $B$ lie in an $\ell_1$ ball. Thus, $\|B\|_2 \to \infty$ if and only if for any sequence of diagonal elements $\|[B_{11}, \ldots, B_{pp}]\|_\infty \to \infty$. On the other hand, by Hadamard's inequality for positive definite matrices [26], we have $2\log\det B \le \sum_k 2\log B_{kk}$. Thus,

$$\|DBM\|_F^2 - 2\log\det B \ge \|DBM\|_F^2 - 2\sum_k \log B_{kk}. \tag{19}$$

We now lower bound $\|DBM\|_F^2$. Consider the following inequality:

$$\|DBM\|_F^2 = \sum_{i,j}[DBM]_{ij}^2 \tag{20}$$

$$= \sum_{i,j,k,l}(D_{ik}B_{kl}M_{lj})^2 \tag{21}$$

$$\ge \sum_{i,j,k=l}(D_{ik}B_{kk}M_{kj})^2 \tag{22}$$

$$= \sum_k (B_{kk})^2 \sum_{i,j}(D_{ik})^2(M_{kj})^2. \tag{23}$$

The inequality in (22) follows because the off-diagonal elements of the matrix $B$ are non-negative (i.e., $\sum_{i,j,k\ne l}(D_{ik}B_{kl}M_{lj})^2 \ge 0$). Rewriting equation (19) using the lower bound from equation (23), we have

$$\|DBM\|_F^2 - 2\log\det B \ge \sum_k \left[(B_{kk})^2 \sum_{i,j}(D_{ik})^2(M_{kj})^2 - 2\log B_{kk}\right] \tag{24}$$

If the term $\sum_{ij}(D_{ik})^2(M_{kj})^2 > 0$, then for any sequence $\|[B_{11}, \ldots, B_{pp}]\|_\infty \to \infty$, the quadratic term $(B_{kk})^2$ in the right hand side of the lower bound in equation (24) dominates the logarithmic term $\log B_{kk}$ for all $k$. Therefore the objective function $CO(B) = \|DBM\|_F^2 - 2\log\det B \ge \infty$ as $\|[B_{11}, \ldots, B_{pp}]\|_\infty \to \infty$. This implies that $CO(B)$ is coercive and a unique minima exists. Now it remains to show that the term $\sum_{ij}(D_{ik})^2(M_{kj})^2 > 0$.

One one hand, by assumption, the diagonal elements of the sample covariance matrix $S$ are strictly positive. On the other hand, $S = M^2$, where $M$ is the unique positive semi-definite square root. Thus,

$$S_{kk} = \sum_j (M_{kj})^2 > 0. \tag{25}$$

By the same logic, the following inequality holds for $D^2 = \Sigma_X^{-1}$, where $\Sigma_X$ is positive definite matrix:

$$[\Sigma_X^{-1}]_{kk} = \sum_i (D_{ik})^2 > 0. \tag{26}$$

Multiplying terms in (25) and (26), we have that $\sum_{ij} (D_{ik})^2 (M_{kj})^2 > 0$.

$\square$

We derive sufficient conditions under which the PDW construction (defined in Section 3.2) succeeds.

**Lemma 2.** *(Sufficient conditions for strict dual feasibility) Let the regularization parameter $\lambda_n > 0$ and $\alpha$ be defined as in* **[A1]**. *Suppose the following holds*

$$\max \left\{ \left\| \Gamma(D^2 \Delta) + \Gamma(D^2 B^*) \right\|_\infty \|W\|_\infty, \|R(\Delta)\|_\infty, \left\| \Gamma(D^2 \Delta) \right\|_\infty \|\Theta^{*-1}\|_\infty \right\} \leq \frac{\lambda_n \alpha}{24}. \tag{27}$$

*Then the dual vector $\widetilde{Z}_{E^c}$ satisfies $\|\widetilde{Z}_{E^c}\|_\infty < 1$, and hence, $\widetilde{B} = \widehat{B}$.*

*Proof.* We begin by obtaining a suitable expression for $\widetilde{Z}_{E^c}$ using the zero-subgradient condition of the the restricted $\ell_1$ regularized log-det problem defined in (6):

$$\widetilde{B} = \underset{B = B^T, B \succ 0, B_{E^c} = 0}{\arg\min} \left[ \text{Tr}(DBSBD) - \log\det(B^2) + \lambda_n \|B\|_{1,\text{off}} \right]. \tag{28}$$

The zero-subgradient of the restricted problem is given by

$$2D^2 \widetilde{B} S - 2\widetilde{B}^{-1} + \lambda_n \widetilde{Z} = 0, \tag{29}$$

where $\widetilde{B}$ is the primal solution given by (28) and $\widetilde{Z} \in \partial \|B\|_{1,\text{off}}$ is the optimal dual. Recall that $\Delta = \widetilde{B} - B^*$ and $W = S - \Theta^{*-1}$ and notice the following chain of identities:

$$
\begin{aligned}
2(D^2 \widetilde{B} S - \widetilde{B}^{-1}) + \lambda_n \widetilde{Z} &= 2(D^2 \widetilde{B} S - D^2 B^* S + D^2 B^* S - \widetilde{B}^{-1}) + \lambda_n \widetilde{Z} \\
&= 2(D^2 \Delta S + D^2 B^* S - \widetilde{B}^{-1}) + \lambda_n \widetilde{Z} \\
&= 2(D^2 \Delta W + D^2 B^* W + D^2 \Delta \Theta^{*-1} + D^2 B^* \Theta^{*-1} - \widetilde{B}^{-1}) + \lambda_n \widetilde{Z}.
\end{aligned}
$$

On the other hand, by definition, $\Theta^{*-1} = B^{*-1} \Sigma_X B^{*-1}$ and $(D^2)^{-1} = \Sigma_X$. Thus, $D^2 B^* \Theta^{*-1} = B^{*-1}$. Substituting these expressions in the zero-subgradient condition yields the following:

$$D^2 \Delta W + D^2 B^* W + D^2 \Delta \Theta^{*-1} + B^{*-1} - \widetilde{B}^{-1} + \lambda_n' \widetilde{Z} = 0, \tag{30}$$

where $\lambda_n' = 0.5\lambda_n$. By adding and subtracting $B^{*-1} \Delta B^{*-1}$ to the preceding equality and followed by some algebraic manipulations give us

$$B^{*-1} \Delta B^{*-1} + D^2 \Delta W + D^2 B^* W + D^2 \Delta \Theta^{*-1} - R(\Delta) + \lambda_n' \widetilde{Z} = 0, \tag{31}$$

where $R(\Delta) = \widetilde{B}^{-1} - B^{*-1} - B^{*-1} \Delta B^{*-1}$.

We now vectorize (30). We use $\text{vec}(A)$ or $\bar{A}$ to denote the $p^2$-vector formed by stacking the columns of $A$ and use $\Gamma(A) = (I \otimes A)$ to denote the Kronecker product of $A$ with the identity matrix $I$. By applying vec() operator on both sides of (30) it follows that

$$\text{vec}(B^{*-1} \Delta B^{*-1} + D^2 \Delta W + D^2 B^* W + D^2 \Delta \Theta^{*-1} - R(\Delta) + \lambda_n' \widetilde{Z}) = 0. \tag{32}$$

Using the standard Kronecker matrix product rules [31], we have $\text{vec}(B^{*-1} \Delta B^{*-1}) = \Gamma^* \bar{\Delta}$, where $\Gamma^* = B^{*-1} \otimes B^{*-1}$ and $\text{vec}((D^2 \Delta)W) = \Gamma(D^2 \Delta)\bar{W}$; $\Gamma(D^2 \Delta) = I \otimes D^2 \Delta$; and $I$ is the $p \times p$ identity matrix. By substituting these observations in (32), we note that

$$\Gamma^* \bar{\Delta} + \Gamma(D^2 \Delta)\bar{W} + \Gamma(D^2 B^*)\bar{W} + \Gamma(D^2 \Delta)\overline{\Theta^{*-1}} - \overline{R(\Delta)} + \lambda_n' \bar{\bar{Z}} = 0. \tag{33}$$

For compactness, we suppress $\Delta$ notation in $R(\Delta)$. Recall that the $E$ is the augmented set defined as $E := \{\mathcal{E}(B^*) \cup (1,1) \ldots \cup (p,p)\}$, where $\mathcal{E}$ is the edge set of the network and $E^c$ is the complement of the set $E$. Recall that we use the notation $A_E$ to denote the sub-matrix of $A$ containing all

elements $A_{ij}$ such that $(i,j) \in E$. We partition the preceding linear equations into two separate linear equations corresponding to the sets $E$ and $E^c$ as

$$\Gamma_{EE}^* \overline{\Delta}_E + \left(\Gamma_{EE}(D^2\Delta) + \Gamma_{EE}(D^2 B^*)\right)\overline{W}_E + \Gamma_{EE}(D^2\Delta)\overline{\Theta_E^{*-1}} - \overline{R}_E + \lambda_n'\overline{\widetilde{Z}}_E = 0,$$
(34)

$$\Gamma_{E^c E}^* \overline{\Delta}_E + \left(\Gamma_{E^c E}(D^2\Delta) + \Gamma_{E^c E}(D^2 B^*)\right)\overline{W}_{E^c} + \Gamma_{E^c E}(D^2\Delta)\overline{\Theta_{E^c}^{*-1}} - \overline{R}_{E^c} + \lambda_n'\overline{\widetilde{Z}}_{E^c} = 0.$$
(35)

From (34), we can solve for $\overline{\Delta}_E$ as

$$\overline{\Delta}_E = (\Gamma_{EE}^*)^{-1}\underbrace{\left[-\left(\left(\Gamma_{EE}(D^2\Delta) + \Gamma_{EE}(D^2 B^*)\right)\overline{W}_E + \Gamma_{EE}(D^2\Delta)\overline{\Theta_E^{*-1}}\right) + \overline{R}_E - \lambda_n'\overline{\widetilde{Z}}_E\right]}_{\triangleq M}.$$
(36)

Substituting $\overline{\Delta}_E$ given by (36) in (35) gives us

$$\Gamma_{E^c E}^*(\Gamma_{EE}^*)^{-1}M + \left(\Gamma_{E^c E}(D^2\Delta) + \Gamma_{E^c E}(D^2 B^*)\right)\overline{W}_{E^c}\Gamma_{E^c E}(D^2\Delta)\overline{\Theta_{E^c}^{*-1}} - \overline{R}_{E^c} + \lambda_n'\overline{\widetilde{Z}}_{E^c} = 0.$$
(37)

From which we can solve for the vectorized dual $\overline{\widetilde{Z}}_{E^c}$ as

$$\lambda_n'\overline{\widetilde{Z}}_{E^c} = -\Gamma_{E^c E}^*(\Gamma_{EE}^*)^{-1}M - \left(\Gamma_{E^c E}(D^2\Delta) + \Gamma_{E^c E}(D^2 B^*)\right)\overline{W}_{E^c} - \Gamma_{E^c E}(D^2\Delta)\overline{\Theta_{E^c}^{*-1}} + \overline{R}_{E^c}.$$
(38)

Taking the element-wise $\ell_\infty$ norm on both sides of the preceding equality gives us

$$\|\overline{\widetilde{Z}}_{E^c}\|_\infty \leq \frac{1}{\lambda_n'}\left|\!\left|\!\left|\Gamma_{E^c E}^*(\Gamma_{EE}^*)^{-1}\right|\!\right|\!\right|_\infty\|M\|_\infty + \frac{1}{\lambda_n'}\left|\!\left|\!\left|\Gamma_{E^c E}(D^2\Delta)\right|\!\right|\!\right|_\infty\|\overline{\Theta_{E^c}^{*-1}}\|_\infty$$
$$+ \frac{1}{\lambda_n'}\left|\!\left|\!\left|\Gamma_{E^c E}(D^2\Delta) + \Gamma_{E^c E}(D^2 B^*)\right|\!\right|\!\right|_\infty\|\overline{W}_{E^c}\|_\infty + \frac{1}{\lambda_n'}\|\!|\overline{R}_{E^c}|\!\|_\infty.$$
(39)

We invoke the mutual incoherence condition in (10) to bound $\left|\!\left|\!\left|\Gamma_{E^c E}^*(\Gamma_{EE}^*)^{-1}\right|\!\right|\!\right|_\infty \leq (1-\alpha)$ and since $\|A_{E^c}\|_\infty \leq \|A\|_\infty$ for any matrix $A$, we get

$$\|\overline{\widetilde{Z}}_{E^c}\|_\infty \leq \frac{1-\alpha}{\lambda_n'}\|M\|_\infty + \frac{1}{\lambda_n'}\left|\!\left|\!\left|\Gamma(D^2\Delta)\right|\!\right|\!\right|_\infty\|\Theta^{*-1}\|_\infty$$
$$+ \frac{1}{\lambda_n'}\left[\left|\!\left|\!\left|\Gamma(D^2\Delta) + \Gamma(D^2 B^*)\right|\!\right|\!\right|_\infty\|W\|_\infty + \|\!|R|\!\|_\infty\right].$$
(40)

We bound $\|M\|_\infty$ by taking the element-wise $\ell_\infty$ norm of $M$ in (36) and followed by applying sub-multiplicative norm inequalites. Thus,

$$\|M\|_\infty \leq \left|\!\left|\!\left|\Gamma_{EE}(D^2\Delta) + \Gamma_{EE}(D^2 B^*)\right|\!\right|\!\right|_\infty\|W_E\|_\infty + \left|\!\left|\!\left|\Gamma_{EE}(D^2\Delta)\right|\!\right|\!\right|_\infty\|\Theta_E^{*-1}\|_\infty$$
$$+ \|\!|R_E|\!\|_\infty + \lambda_n'\|\widetilde{Z}_E\|_\infty.$$
(41)

Because $\widetilde{Z}_E$ is the sub-vector of the vectorized optimal dual $\widetilde{Z}$, it follows that $\|\widetilde{Z}_E\|_\infty \leq 1$. Thus,

$$\|M\|_\infty \leq \underbrace{\left[\left|\!\left|\!\left|\Gamma(D^2\Delta) + \Gamma(D^2 B^*)\right|\!\right|\!\right|_\infty\|W\|_\infty + \left|\!\left|\!\left|\Gamma(D^2\Delta)\right|\!\right|\!\right|_\infty\|\Theta^{*-1}\|_\infty + \|\!|R|\!\|_\infty\right]}_{\triangleq H} + \lambda_n'.$$
(42)

On the other hand, from (42), we have $H \leq \lambda_n'\alpha/4$. Putting together the pieces, from (42) and (40) we conclude that

$$\|\overline{\widetilde{Z}}_{E^c}\|_\infty \leq (1-\alpha) + \frac{1-\alpha}{\lambda_n'}H + \frac{1}{\lambda_n'}H$$
(43)

$$= (1-\alpha) + \frac{2-\alpha}{\lambda_n'}H$$
(44)

$$\leq (1-\alpha) + \frac{2-\alpha}{\lambda_n'}\left(\frac{\lambda_n'\alpha}{4}\right)$$
(45)

$$\leq (1-\alpha) + \frac{\alpha}{2} < 1.$$
(46)

**Remark** For comparison, consider the strict dual feasibility conditions in [48, Lemma 4]. Here, the maximum is on the noise deviation $\|W\|_\infty$ and the remainder term $\|R(\Delta)\|_\infty$. Instead, in our case, the maximum is taken over several other quantities not just $\|W\|_\infty$ and $\|R(\Delta)\|_\infty$ (see (2)). □

The following lemma shows that the remainder term $R(\Delta)$ is bounded if $\Delta$ is bounded. The proof is adapted from [48], where a similar result is derived using matrix expansion techniques. We use this lemma in the proof of our main result (see Theorem 1 and Theorem 2) to show that with sufficient number of samples $R(\Delta) \leq \alpha\lambda_n/24$.

**Lemma 3.** *(Control of reminder) Suppose that the element-wise $\ell_\infty$-bound $\|\Delta\|_\infty \leq \frac{1}{3\nu_{B^{*-1}}d}$ holds, then the matrix $Q = \sum_{k=0}^\infty (-1)^k (B^{*-1}\Delta)^k$ satisfies the bound $\nu_{Q^T} \leq \frac{3}{2}$ and the matrix $R(\Delta) = B^{*-1}\Delta B^{*-1}\Delta Q B^{*-1}$ has the element-wise $\ell_\infty$-norm bounded as*

$$\|R(\Delta)\|_\infty \leq \frac{3}{2}d\|\Delta\|_\infty^2 \nu_{B^{*-1}}^3. \tag{47}$$

We show that for a specific choice of radius $r$, the distortion $\Delta = \widetilde{B} - B^*$ lies in a ball of radius $r$.

**Lemma 4.** *(Control of $\Delta$) Let*

$$r \triangleq 4\nu_{\Gamma^{*-1}}\left[\nu_{D^2}\nu_{B^*}\|W\|_\infty + 0.5\lambda_n\right] \leq \min\left\{\frac{1}{3\nu_{B^{*-1}}d}, \frac{1}{6\nu_{\Gamma^{*-1}}\nu_{B^{*-1}}^3 d}\right\}.$$

*Then we have the element-wise $\ell_\infty$ bound $\|\Delta\|_\infty = \|\widetilde{B} - B^*\|_\infty \leq r$.*

*Proof.* We adopt the proof technique in [48, Lemma 6]. We use the notation $A_E$ or $[A]_E$ to denote the sub-matrix of $A$ containing all elements $A_{ij}$ such that $(i,j) \in E$. Let $G(\widetilde{B}_E)$ be the zero sub-gradient condition of the restricted $\ell_1$ log-det problem in (12):

$$G(\widetilde{B}_E) = \left[D^2\widetilde{B}S - \widetilde{B}^{-1} + \lambda_n'\widetilde{Z}\right]_E = 0. \tag{48}$$

where $\lambda_n' = 0.5\lambda_n$. Let $\overline{G}$ denote the vectorized form of $G$. Recall that $\Delta = \widetilde{B} - B^* = \widetilde{B}_E - B_E^* \triangleq \Delta_E$. The second equality follows from PDW construction and the constraint in the restricted convex program in (6). To establish $\|\Delta\|_\infty \leq r$, we show that $\Delta_E$ lies inside the ball $\mathbb{B}_r = \{\bar{A}_E \in \mathbb{R}^{|E|} : \|A\|_\infty \leq r\}$, where $\bar{A}_E = \text{vec}(A_E)$, using a contraction property of the continuous map:

$$F(\overline{\Delta}_E) \triangleq -(\Gamma_{EE}^*)^{-1}\left(\overline{G}(\Delta_E + B_E^*)\right) + \overline{\Delta}_E, \tag{49}$$

where we used the fact that $\widetilde{B}_{E^c} = \widetilde{B}_{E^c}^* = 0$.

Suppose that $F(\cdot)$ is a contraction on $\mathbb{B}_r$, i.e., $F(\mathbb{B}_r) \subseteq \mathbb{B}_r$. Then by Brower's fixed point theorem [29], it readily follows that there exists a $C \in \mathbb{B}_r$ such that $F(C) = C$. Finally, $C = \overline{\Delta}_E$ because (i) $\widetilde{B}$ that satisfies $\overline{G}(\widetilde{B}) = 0$ is unique (see Lemma 1) and (ii) $F(\overline{\Delta}_E) = \overline{\Delta}_E$ if and only if $\overline{G}(\cdot) = 0$, Hence, $\overline{\Delta}_E \in \mathbb{B}_r$ is the unique fixed point of $F(\cdot)$ in (49). Consequently, $\|\Delta_E\|_\infty \leq r$.

It remains to show that $F(\cdot)$ is a contraction. Let $\Delta' \in \mathbb{R}^{p \times p}$ be a zero padded matrix on $E^c$ such that $\overline{\Delta}'_E \in \mathbb{B}_r$. Then $F(\overline{\Delta}'_E)$ can be expanded in terms of $\Delta'$ as

$$\begin{aligned}
F(\overline{\Delta}'_E) &= -(\Gamma_{EE}^*)^{-1}\left(\overline{G}(\Delta'_E + B_E^*)\right) + \overline{\Delta}'_E \\
&= -(\Gamma_{EE}^*)^{-1}\left[\text{vec}([D^2(\Delta' + B^*)S]_E - (\Delta' + B^*)_E^{-1} + \lambda_n'\widetilde{Z}_E) + \Gamma_{EE}^*\overline{\Delta}'_E\right].
\end{aligned} \tag{50}$$

Adding and subtracting $\Theta^{*-1}$ and $B_E^{*-1}$ to the preceding equality yields us

$$\begin{aligned}
F(\overline{\Delta}'_E) = &-(\Gamma_{EE}^*)^{-1}\left[\text{vec}\left([D^2(\Delta' + B^*)W]_E + [D^2(\Delta' + B^*)\Theta^{*-1}]_E + \lambda_n'\widetilde{Z}_E - B_E^{*-1}\right)\right] \\
&-(\Gamma_{EE}^*)^{-1}\left[-\text{vec}\left((\Delta' + B^*)^{-1} - B_E^{*-1}\right) + \Gamma_{EE}^*\overline{\Delta}'_E\right].
\end{aligned} \tag{51}$$

The last vec () term can be even simplified as

$$
\begin{aligned}
\text{vec}\left((\Delta' + B^*)^{-1} - B^{*-1}\right) + \Gamma^*\Delta' &= \text{vec}\left((\Delta' + B^*)^{-1} - B^{*-1} + (B^{*-1}\Delta'B^{*-1})\right) \\
&= \text{vec}(R(\Delta')).
\end{aligned}
\tag{52}
$$

Substituting this observation in (51) and rearranging the terms gives us

$$
F(\overline{\Delta}'_E) = - \underbrace{(\Gamma^*_{EE})^{-1} \text{vec}\left[D^2 B^* W \lambda'_n \widetilde{Z}\right]_E}_{\triangleq T_1} - \underbrace{(\Gamma^*_{EE})^{-1} \text{vec}\left[D^2 \Delta' W\right]_E}_{\triangleq T_2}
$$
$$
\tag{53}
$$
$$
- \underbrace{(\Gamma^*_{EE})^{-1}\left[\overline{R}(\Delta')\right]_E}_{\triangleq T_3} - \underbrace{(\Gamma^*_{EE})^{-1} \text{vec}\left[D^2(\Delta' + B^*)\Theta^{*-1} - B^{*-1}\right]_E}_{\triangleq T_4}.
$$

We now show that $\|F(\overline{\Delta}'_E)\|_\infty \leq r$ by bounding $\ell_\infty$ norms of terms $(T_1)$-$(T_4)$. Recall that $\nu_A = \|\!|A|\!\|_\infty \triangleq \max_{j=1,\dots,p} \sum_{j=1}^p |A_{ij}|$ and it is sub-multiplicative; that is $\|\!|AB|\!\|_\infty \leq \|\!|A|\!\|_\infty \|\!|B|\!\|_\infty$. Notice that this not the case with the max norm ($\ell_\infty$).

(i) *upper bound on* $\|T_1\|_\infty$: Consider the following chain of inequalities.

$$
\begin{aligned}
\|T_1\|_\infty &\leq \left\|\!\left|\Gamma^{*-1}\right|\!\right\|_\infty \left\|\text{vec}(D^2 B^* W + \lambda'_n \widetilde{Z})\right\|_\infty \\
&= \left\|\!\left|\Gamma^{*-1}\right|\!\right\|_\infty \left\|\Gamma(D^2 B^*)\overline{W} + \lambda'_n \overline{\widetilde{Z}}\right\|_\infty \\
&\overset{(a)}{\leq} \left\|\!\left|\Gamma^{*-1}\right|\!\right\|_\infty \left[\|\!|\Gamma(D^2 B^*)|\!\|_\infty \|W\|_\infty + \lambda'_n\right] \\
&\overset{(b)}{\leq} \nu_{\Gamma^{*-1}}\left[\nu_{D^2}\nu_{B^*}\|W\|_\infty + \lambda'_n\right] \overset{(c)}{\leq} \frac{r}{4}
\end{aligned}
\tag{54}
$$

where (a) follows because $\|\overline{\widetilde{Z}}\|_\infty \leq 1$ (see Lemma 1); (b) follows because $\Gamma(D^2 B^*) = (I \otimes D^2 B^*)$, and hence, $\|\!|\Gamma(D^2 B^*)|\!\|_\infty = \|\!|D^2 B^*|\!\|_\infty \leq \|\!|D^2|\!\|_\infty \|\!|B^*|\!\|_\infty = \nu_{D^2}\nu_{B^*}$; and finally, (c) follows from definition of the radius $r$ in Lemma 4.

(ii) *upper bound on* $\|T_2\|_\infty$: For $T_2$ in (53), consider the following chain of inequalities.

$$
\begin{aligned}
\|T_2\|_\infty &\leq \nu_{\Gamma^{*-1}}\left[\|\!|\Gamma(D^2 \Delta)|\!\|_\infty \|W\|_\infty\right] \\
&\leq \nu_{\Gamma^{*-1}}\nu_{D^2}\|\!|\Delta'|\!\|_\infty \|W\|_\infty \\
&\overset{(a)}{\leq} \nu_{\Gamma^{*-1}}\nu_{D^2}d\|\Delta'\|_\infty \|W\|_\infty \\
&\overset{(b)}{\leq} \nu_{\Gamma^{*-1}}\nu_{D^2}d\|\Delta'\|_\infty \left(\frac{r}{4\nu_{\Gamma^{*-1}}\nu_{D^2}\nu_{B^*}}\right) \\
&\overset{(c)}{\leq} d\left(\frac{1}{3d\nu_{B^{*-1}}}\right)\left(\frac{r}{4\nu_{B^*}}\right) \overset{(d)}{\leq} \frac{r}{4},
\end{aligned}
\tag{55}
$$

where (a) follows because by construction $\Delta'$ has at-most $d$ non-zeros in every row and that $\|\!|\Delta'|\!\|_\infty \leq d\|\Delta'\|_\infty$; (b) follows from the choice of $r = 4\nu_{\Gamma^{*-1}}(\nu_{D^2}\nu_{B^*}\|W\|_\infty + \lambda'_n)$ in Lemma 4, which is lower bounded by $4\nu_{\Gamma^{*-1}}\nu_{D^2}\nu_{B^*}\|W\|_\infty$, for all $\lambda'_n \geq 0$. Thus, $\|W\|_\infty \leq r/(4\nu_{\Gamma^{*-1}}\nu_{D^2}\nu_{B^*})$; (c) follows because $\Delta'$ is a zero-padded matrix of $\Delta$. Hence $\|\Delta\| = \|\Delta'\|_\infty \leq r$, which can be upper bounded by $1/(3d\nu_{B^{*-1}})$ in light of the hypothesis in Lemma 4; and finally, (d) follows because $\nu_{B^*}\nu_{B^{*-1}} \geq 1$.

(iii) *upper bound on* $\|T_3\|_\infty$: For $T_3$ in (53), consider the following chain of inequalities.

$$
\|T_3\|_\infty \leq \nu_{\Gamma^{*-1}}\|R(\Delta')\|_\infty
\tag{56}
$$
$$
\overset{(a)}{\leq} \frac{3}{2}d\nu_{\Gamma^{*-1}}\nu^3_{B^{*-1}}\|\Delta'\|^2_\infty
\tag{57}
$$
$$
\overset{(b)}{\leq} \frac{3}{2}d\nu_{\Gamma^{*-1}}\nu^3_{B^{*-1}}r(r) \overset{(c)}{\leq} \frac{r}{4},
\tag{58}
$$

where (a) follows because Lemma 3 guarantees that $\|R(\Delta')\|_\infty \leq (3/2)d\nu_{B^{*-1}}^3\|\Delta'\|_\infty^2$ whenever $\|\Delta'\|_\infty \leq 1/(3d\nu_{B^{*-1}})$. The latter inequality is a consequence of the hypothesis in Lemma 4; (b) is true because by construction $\Delta' \in \mathbb{B}_r$, and hence, $\|\Delta'\|_\infty \leq r$; (c) follows by invoking the hypothesis in Lemma 4, where $r$ satisfies $r \leq 1/(6d\nu_{\Gamma^{*-1}}\nu_{B^{*-1}}^3)$.

(iv) *upper bound on $\|T_4\|_\infty$*: The expression of $T_4$ in (53) can be simplified as

$$
\begin{aligned}
T_4 &= -(\Gamma_{EE}^*)^{-1} \operatorname{vec}\left(\left[D^2(\Delta' + B^*)\Theta^{*-1} - B^{*-1}\right]_E\right)\\
&= -(\Gamma_{EE}^*)^{-1} \operatorname{vec}\left(\left[D^2\Delta'\Theta^{*-1} + D^2 B^*\Theta^{*-1} - B^{*-1}\right]_E\right)\\
&= -(\Gamma_{EE}^*)^{-1} \operatorname{vec}\left(\left[D^2\Delta'\Theta^{*-1}\right]_E\right).
\end{aligned}
\tag{59}
$$

The last equality follows by observing that $D^2 B^*\Theta^{*-1} = B^{*-1}$. This can be verified by plugging $\Theta^{*-1} = B^{*-1}\Sigma_X B^{*-1}$ and $\Sigma_X = (D^2)^{-1}$ in $D^2 B^*\Theta^{*-1}$ and simplifying the resulting expression. By taking the $\ell_\infty$ bound on the both sides of (59) gives us

$$
\|T_4\|_\infty \leq \nu_{\Gamma^{*-1}}\left\|\left|\Gamma(D^2\Delta')\right|\right\|_\infty\|\Theta^{*-1}\|_\infty
\tag{60}
$$

$$
\leq \nu_{\Gamma^{*-1}}\nu_{D^2}d\|\Delta'\|_\infty\|\Theta^{*-1}\|_\infty
\tag{61}
$$

$$
\leq \nu_{\Gamma^{*-1}}\nu_{D^2}rd\|\Theta^{*-1}\|_\infty \overset{(a)}{\leq} \frac{r}{4},
\tag{62}
$$

where (a) follows by invoking the assumption in (11), and noting that $\|\Theta^{*-1}\|_\infty \leq 1/(4\nu_{\Gamma^{*-1}}\nu_{D^2}d)$.

Putting together the pieces, from the above calculations, we note that

$$
\|F(\Delta'_E)\|_\infty \leq \|T_1\|_\infty + \|T_2\|_\infty + \|T_3\|_\infty + \|T_4\|_\infty \leq r.
\tag{63}
$$

is a contraction as claimed. This concludes the proof. $\qquad\square$

We borrow the following notion of tail conditions as defined in [48] to characterize the distribution. We us this characterization to prove our master lemma A.6.

**Definition A.3.** *(Tail condition, [48]) The random vector $Y$ satisfies the tail condition $\mathcal{T}(f, v_*)$ if there exist a constant $v_* > 0$ and a function $f : \mathbb{N} \times (0, \infty)$ such that for any $i, j \in [p]$ and $\delta \in (0, 1/v_*)$:*

$$
\mathbb{P}\left[|S_{ij} - \Sigma_{ij}^*| \geq \delta\right] \leq \frac{1}{f(n, \delta)}.
\tag{64}
$$

*Furthermore, $f(n, \delta)$ is monotonically increasing in $n$ (or $\delta$) for fixed $\delta$ (or $n$).*

Both the exponential-type tail $f(n, \delta) = \exp(cn\delta^a)$ and the polynomial-type tail $f(n, \delta) = cn^m\delta 2m$, where $m$ is an integer and $c, a > 0$, satisfy the monontone property in Definition A.3. The following inverse functions associated with $f(n, \delta)$ are needed to prove our sample complexity result:

$$
n_f(\delta, p^\tau) := \max\{n | f(n, \delta) \leq p^\tau\} \text{ and } \delta_f(n, p^\tau) := \max\{\delta | f(n, \delta) \leq p^\tau\}.
\tag{65}
$$

Both the functions are well-defined due to the to the monotonicity property of $f(n, \delta)$. Further, if $n > n_f(\delta, p^\tau)$ for some $\delta > 0$ implies that $\delta \geq \delta_f(n, p^\tau)$.

The following result presents an exponential-type tail bound for sub-Gaussian random vectors.

*Lemma A.5.* *(Sub-Gaussian tail condition, [48]) Consider a zero-mean random vector $(Y_1, \ldots, Y_p)$ with covariance $\Sigma^*$ such that each $Y_i/\sqrt{\Sigma_{ii}^*}$ is sub-Gaussian with parameter $\sigma$. Given $n$ i.i.d samples, the sample covariance matrix $S$ satisfies the tail bound*

$$
\mathbb{P}\left[|S_{ij} - \Sigma_{ij}^*| > \delta\right] \leq 4\exp\left\{-\frac{n\delta^2}{128(1 + 4\sigma^2)^2\max_i(\Sigma_{ii}^*)^2}\right\},
\tag{66}
$$

*for all $\delta \in (0, 8(1 + 4\sigma^2)\max_i(\Sigma_{ii}^*))$.*

Let $W_{ij} = S_{ij} - \Sigma^*_{ij}$, where $\Sigma_* = \Theta^{*-1}$. This difference quantity, which signifies the amount of noise in the data, plays a key role in bounding the error term $\|\widehat{B} - B^*\|_\infty$. We later show that if $W_{ij}$ is small, then we can guarantee that our estimator $\widehat{B}$ is close to $B^*$ in the element-wise $\ell_\infty$ −norm.

By taking a union bound over all entries of $|W_{ij}|$, from Lemma A.5, it follows that

$$\mathbb{P}\left[\|W\|_\infty \geq \delta_f(n, p^\tau)\right] \leq \frac{p^2}{f(n, \delta_f(n, p^\tau))} = \frac{1}{p^{\tau-2}}, \tag{67}$$

for some $\tau > 2$. The above bound gives an explicit control on the noise term.

We now state and prove our master lemma which gives support recovery guarantees and $\ell_\infty$ norm bounds for our estimator $\widehat{B}$ for distributions satisfying tail condition $\mathcal{T}(f, v_*)$ in Definition A.3.

**Lemma A.6.** *(Master lemma) Consider a distribution satisfying the incoherence assumption with parameter $\alpha \in (0, 1]$ and the tail condition $\mathcal{T}(f, v_*)$. Let $\widehat{B}$ be the unique solution of the log-determinant problem in (9) with $\lambda_n = 2\nu_{D^2}\nu_{B^*}\delta_f(n, p^\tau)$ for some $\tau > 2$. Then if the sample size is lower bounded as*

$$n > n_f(1/\max\{v_*, 24d\nu_{D^2}\nu_{B^*}\max\{\nu_{\Gamma^{*-1}}\nu_{B^{*-1}}, 2\nu^2_{\Gamma^{*-1}}\nu^3_{B^{*-1}}, 2\alpha^{-1}d^{-1}\}\}, p^\tau), \tag{68}$$

*then with probability greater than $1 - \frac{1}{p^{\tau-2}}$, the estimate $\widehat{B}$ recovers the sparsity structure of $B^*$ ie. $(\widehat{B}_{E^c} = B^*_{E^c})$. Furthermore $\widehat{B}$ satisfies the $\ell_\infty$ bound $\|\widehat{B} - B^*\|_\infty \leq 8\nu_{\Gamma^{*-1}}\nu_{D^2}\nu_{B^*}\delta_f(n, p^\tau)$.*

*Proof.* We first show that the Primal Dual Witness (PDW) construction (see sec 3.3) succeeds with the probability stated in the lemma. This amounts to showing that the inequality in (8) holds with the required probability. To this aim, let $\mathcal{A}$ denote the event that $\|W\|_\infty \leq \delta_f(n, p^\tau)$. We have previously shown in (67) that $\mathbb{P}[\mathcal{A}] \geq 1 - 1/p^{\tau-2}$. Conditioned on the event $\mathcal{A}$, we show that the inequality in (8) is satisfied.

From Lemma 4, we have

$$r = 4\nu_{\Gamma^{*-1}}\left[\nu_{D^2}\nu_{B^*}\|W\|_\infty + 0.5\lambda_n\right], \tag{69}$$

substituting for $\lambda_n = 2\nu_{D^2}\nu_{B^*}\delta_f(n, p^\tau)$ as given in the assumption, we get

$$r \leq 8\nu_{\Gamma^{*-1}}\nu_{D^2}\nu_{B^*}\delta_f(n, p^\tau). \tag{70}$$

From assumption on the sample size $n$ in (68) and the monotonicity property (65) we have $0.5\lambda_n = \nu_{D^2}\nu_{B^*}\delta_f(n, p^\tau) \leq \alpha/48$, which implies that $\lambda_n < 1$. We also set $\delta_f(n, p^\tau) \leq \lambda_n$. Similarly from (65) and (68) we have $r \leq 8\nu_{\Gamma^{*-1}}\nu_{D^2}\nu_{B^*}\delta_f(n, p^\tau) \leq \min\{1/(3d\nu_{B^{*-1}}), 1(6d\nu_{\Gamma^{*-1}}\nu^3_{B^{*-1}})\}$. Therefore the assumption in Lemma 4 is satisfied resulting in

$$\|\Delta\|_\infty \leq r \leq \min\left[\frac{1}{3d\nu_{B^{*-1}}}, \frac{1}{6d\nu_{\Gamma^{*-1}}\nu^3_{B^{*-1}}}\right]. \tag{71}$$

Define $\delta_f \triangleq \delta_f(n, p^\tau)$. We show that the every component in the max term of (8) are bounded by $\alpha\lambda_n/24$. We begin with the first component:

$$\left\|\left|\Gamma(D^2\Delta) + \Gamma(D^2B^*)\right|\right\|_\infty\|W\|_\infty \leq \left[\left\|\left|D^2\Delta + D^2B^*\right|\right\|_\infty\right]\delta_f \tag{72}$$

$$\leq \left[\left\|\left|D^2\Delta\right|\right\|_\infty + \left\|\left|D^2B^*\right|\right\|_\infty\right]\delta_f \tag{73}$$

$$\leq \left[\nu_{D^2}d\|\Delta\|_\infty + \nu_{D^2}\nu_{B^*}\right]\frac{\alpha\lambda_n}{48\nu_{D^2}\nu_{B^*}} \tag{74}$$

$$\overset{(a)}{\leq} \left[1 + \frac{1}{3\nu_{B^*}\nu_{B^{*-1}}}\right]\frac{\alpha\lambda_n}{48} \tag{75}$$

$$\overset{(b)}{\leq} \frac{\alpha\lambda_n}{36} \leq \frac{\alpha\lambda_n}{24}. \tag{76}$$

where (a) follows from (71); (b) follows because $\nu_{B^*}\nu_{B^{*-1}} \geq 1$.

We show the second component $\|R(\Delta)\|_\infty \le \alpha\lambda_n/24$. In fact,

$$\|R(\Delta)\|_\infty \overset{(a)}{\le} \frac{3}{2}d\|\Delta\|_\infty^2 \nu_{B^{*-1}}^3 \tag{77}$$

$$\overset{(b)}{\le} \frac{3}{2}dr\nu_{B^{*-1}}^3 r \tag{78}$$

$$\overset{(c)}{\le} \frac{3}{2}d\left[\frac{1}{6d\nu_{\Gamma^{*-1}}\nu_{B^{*-1}}^3}\right]\nu_{B^{*-1}}^3(8\nu_{\Gamma^{*-1}}\nu_{D^2}\nu_{B^*}\delta_f) \tag{79}$$

$$= 2\nu_{D^2}\nu_{B^*}\delta_f \le \frac{\alpha\lambda_n}{24}. \tag{80}$$

where (a) holds because, as shown in (71), $\|\Delta\|_\infty$ satisfies the assumption in Lemma 3; (b) holds because $\|\Delta\|_\infty \le r$; and (c) is a consequence of the inequality in (71).

We show that the third component $\left\|\left|\Gamma(D^2\Delta)\right|\right\|_\infty\|\Theta^{*-1}\|_\infty \le \alpha\lambda_n/24$. In fact,

$$\left\|\left|\Gamma(D^2\Delta)\right|\right\|_\infty\|\Theta^{*-1}\|_\infty = \left\|\left|D^2\Delta\right|\right\|_\infty\|\Theta^{*-1}\|_\infty \tag{81}$$

$$\le \nu_{D^2}d\|\Theta^{*-1}\|_\infty\|\Delta\|_\infty \tag{82}$$

$$\overset{(a)}{\le} \nu_{D^2}d\|\Theta^{*-1}\|_\infty r \tag{83}$$

$$\overset{(b)}{\le} d\nu_{D^2}\left[\frac{1}{4d\nu_{D^2}\nu_{\Gamma^{*-1}}}\right][8\nu_{\Gamma^{*-1}}\nu_{D^2}\nu_{B^*}\delta_f] \tag{84}$$

$$\le 2\nu_{D^2}\nu_{B^*}\left[\frac{\alpha\lambda_n}{48\nu^{D^2}\nu_{B^*}}\right] = \frac{\alpha\lambda_n}{24}. \tag{85}$$

where (a) holds because $\|\Delta\|_\infty \le r$ and (b) follows by invoking the assumption in (11). Since the sufficient conditions for strict dual feasibility are satisfied, the PDW construction succeeds. Therefore $\widehat{B} = \widetilde{B}$. Since by definition $\widetilde{B}_E = B^*_{E^c} = 0$, the estimator $\widehat{B}$ recovers the sparsity structure of $B^*$. Now, since $\Delta = \widehat{B} - B^*$ and $\|\Delta\|_\infty \le 8\nu_{\Gamma^{*-1}}\nu_{D^2}\nu_{B^*}\delta_f(n,p^\tau)$, we have $\|\widehat{B} - \widetilde{B}\|_\infty \le 8\nu_{\Gamma^{*-1}}\nu_{D^2}\nu_{B^*}\delta_f(n,p^\tau)$. $\square$

We use Lemma A.5 and Lemma A.6 to prove our main result for sub-gaussian distributions.

**Theorem 1.** *(Support recovery: Sub-Gaussian) Let $Y = (Y_1,\ldots,Y_p)$ be the node potential vector. Suppose that $Y_i/\sqrt{\Sigma^*_{ii}}$ is sub-Gaussian with parameter $\sigma$ and assumptions [A1-A2] hold. Let the regularization parameter $\lambda_n = C_0\sqrt{\tau(\log 4p)/n}$, where $C_0$ is given below. If the sample size $n > C_1^2 d^2(\tau\log p + \log 4)$, the following hold with probability at least $1 - \frac{1}{p^{\tau-2}}$, for some $\tau > 2$:*

(a) *$\widehat{B}$ exactly recovers the sparsity structure of $B^*$; that is, $\widehat{B}_{E^c} = 0$,*

(b) *$\widehat{B}$ satisfies the element-wise $\ell_\infty$ bound $\|\widehat{B} - B^*\|_\infty \le C_2\sqrt{\frac{\tau\log p + \log 4}{n}}$, and*

(c) *$\widehat{B}$ satisfies sign consistency if $|B^*_{\min}| \ge 2C_2\sqrt{\frac{\tau\log p+4}{n}}$, $B^*_{\min} \triangleq \min_{(i,j)\in\mathcal{E}(B^*)}|B^*_{ij}|$,*

*where $C_1 = 192\sqrt{2}\left[(1+4\sigma^2)\max_i(\Sigma^*_{ii})\nu_{D^2}\nu_{B^*}\right]\max\{\nu_{\Gamma^{*-1}}\nu_{B^{*-1}}, 2\nu_{\Gamma^{*-1}}^2\nu_{B^{*-1}}^3, 2\alpha^{-1}d^{-1}\}$, $C_2 = [64\sqrt{2}(1+4\sigma^2)\max_i(\Sigma^*_{ii})\nu_{\Gamma^{*-1}}\nu_{D^2}\nu_{B^*}]$, and $C_0 = C_2/(4\nu_{\Gamma^{*-1}})$.*

*Proof.* Part (a): From Lemma A.6, we have that if $n > n_f(\delta, p^\tau)$, then $\widehat{B}$ recovers the exact sparsity structure of $B^*$. We compute $n_f(\delta, p^\tau)$. Using the tail bound for sub-gaussian distributions (see Lemma A.5), the decay function $f(n,\delta) = \frac{1}{4}\exp\left\{\frac{n\delta^2}{c_*}\right\}$, where $c_* = 128(1+4\sigma^2)^2\max_i(\Sigma^*_{ii})^2$. From the definition of inverse function and monotonicity of $f(n,\delta)$ in A.3, we have $n_f(\delta, p^\tau) = \frac{c_*\log(4p^\tau)}{\delta^2}$. Substituting for $\delta$ from Lemma A.6, we get

$$n_f(\delta, p^\tau) = C_1^2 d^2(\tau\log p + \log 4). \tag{86}$$

Therefore, from Lemma A.6, if $n > C_1^2 d^2(\tau \log p + \log 4)$, the estimator $\widehat{B}$ recovers the sparsity structure of $B^*$.

Part(b): From Lemma A.5 we compute $\delta$. Using the monotonicity property of $f(n, \delta)$ and setting

$$\delta \triangleq \delta_f(n, p^\tau) = \sqrt{\frac{c_* \log(4p^\tau)}{n}} = 8\sqrt{2}(1 + 4\sigma^2) \max_i(\Sigma_{ii}^*) \sqrt{\frac{\tau \log p + \log 4}{n}}. \qquad (87)$$

Also we have from Lemma A.6 that $\|\widehat{B} - B^*\|_\infty \leq 8\nu_{\Gamma^{*-1}}\nu_{D^2}\nu_{B^*}\delta_f(n, p^\tau)$. Thus,

$$\|\widehat{B} - B^*\|_\infty \leq \underbrace{64\sqrt{2}(1 + 4\sigma^2) \max_i(\Sigma_{ii}^*)\nu_{\Gamma^{*-1}}\nu_{D^2}\nu_{B^*}}_{C_2} \sqrt{\frac{\tau \log p + \log 4}{n}}. \qquad (88)$$

Part(c): We prove the sign consistency of $\widehat{B}$ by contradiction. Let $|B_{\min}^*| \geq 2C_2\sqrt{\frac{\tau \log p + 4}{n}}$ be as in the theorem's hypothesis. Suppose that $\text{sign}(\widehat{B}) \neq \text{sign}(B^*)$. Then, an elementary algebra shows that $\|\widehat{B} - B^*\|_\infty > 2C_2\sqrt{\frac{\tau \log p + 4}{n}}$. This contradicts the bound in part (b). Thus, $\text{sign}(\widehat{B}) \neq \text{sign}(B^*)$. $\qquad \square$

We now show Frobenius and spectral norm consistency for the sub-gaussian distribution. Recall that $\mathcal{E}(B^*) = \{(i, j) : B_{ij}^* \neq 0, \text{for all } i \neq j\}$ is the edge set of $B^*$. Thus, $s = |\mathcal{E}(B^*)|$ is the number of non-zero off-diagonal elements in $B^*$.

**Corollary 1.** *Let $s = |\mathcal{E}(B^*)|$ be the cardinality of $\mathcal{E}(B^*)$. Under the same hypotheses in Theorem 1, with probability greater than $1 - \frac{1}{p^{\tau-2}}$, the estimator $\widehat{B}$ satisfies*

$$\|\widehat{B} - B^*\|_F \leq C_2\sqrt{\frac{(s + p)(\tau \log p + 4)}{n}} \quad and \quad \|\widehat{B} - B^*\|_2 \leq C_2 \min\{d, \sqrt{s + p}\}\sqrt{\frac{\tau \log p + 4}{n}}. \qquad (89)$$

*Proof.* Consider the following inequality:

$$\|\widehat{B} - B^*\|_F^2 = \sum_{i,j}\left(\widehat{B}_{ij} - B_{ij}^*\right)^2 = \sum_i\left(\widehat{B}_{ii} - B_{ii}^*\right)^2 + \sum_{i \neq j}\left(\widehat{B}_{ij} - B_{ij}^*\right)^2 \qquad (90)$$

$$\leq p\|\widehat{B} - B^*\|_\infty^2 + s\|\widehat{B} - B^*\|_\infty^2 \qquad (91)$$

$$= (s + p)\|\widehat{B} - B^*\|_\infty^2, \qquad (92)$$

where the inequality follows because there are at most $p$ non-zero diagonal terms and $s$ non-zero off-diagonal terms in $\widehat{B} - B^*$. The latter fact is a consequence of Theorem 1 (a), which ensures that $\widehat{B}_{E^c} = B_{E^c}^*$ with high probability when $n = \Omega(d^2 \log p)$. We obtain the Frobenius norm bound in (89) by upper bounding $\|\widehat{B} - B^*\|_\infty$ using the result in Theorem 1 (b).

We now spectral norm consistency. From matrix norm equivalence conditions [26], we have

$$\|\widehat{B} - B^*\|_2 \leq \left\|\left\|\widehat{B} - B^*\right\|\right\|_\infty \leq d\|\widehat{B} - B^*\|_\infty \qquad (93)$$

and that

$$\|\widehat{B} - B^*\|_2 \leq \|\widehat{B} - B^*\|_F \leq \sqrt{s + p}\|\widehat{B} - B^*\|_\infty. \qquad (94)$$

These two bounds can be unified into one single bound as

$$\|\widehat{B} - B^*\|_2 \leq \min\{\sqrt{s + p}, d\}\|\widehat{B} - B^*\|_\infty. \qquad (95)$$

This concludes the proof. $\qquad \square$

Next we prove our second main result for random vectors with bounded moments. We need the following standard concentration inequality result.

**Lemma A.7.** *(Tail bounds for random variables with bounded moments, [48]) For a random vector* $(Y_1, \ldots, Y_p)$, *suppose there exists a positive integer* $k$ *and scalar* $M_k \in \mathbb{R}$ *with*

$$\mathbb{E}\left[\frac{Y_i}{\sqrt{\Sigma_{ii}^*}}\right]^{4k} \leq M_k. \tag{96}$$

*Given* $n$ *i.i.d samples, the sample covariance matrix* $S$ *admits the following concentration inequality*

$$\mathbb{P}\left[|S_{ij} - \Sigma_{ij}^*| > \delta\right] \leq \frac{2^{2k}(\max_i \Sigma_{ii}^*)^{2k} C_k(M_k + 1)}{n^k \delta^{2k}}. \tag{97}$$

*where* $C_k \geq 0$ *is a constant depending only on* $k$.

**Theorem 2.** *(Support Recovery: Bounded Moments) Let* $Y = (Y_1, \ldots, Y_p)$ *be the node potential vector. Suppose that* $Y_i/\sqrt{\Sigma_{ii}^*}$ *has bounded moment as in* (96) *and assumptions [A1-A2] hold. Let the regularization parameter* $\lambda_n = C_0 \sqrt{\tau(\log 4p)/n}$, *with* $C_0$ *defined in Theorem 1. If the sample size* $n > C_4 d^2 p^{\tau/k}$, *then with probability more than* $1 - 1/p^{\tau-2}$, *for some* $\tau > 2$, *the following hold:*

(a) $\widehat{B}$ *exactly recovers the sparsity structure of* $B^*$; *that is,* $\widehat{B}_{E^c} = 0$,

(b) $\widehat{B}$ *satisfies the element-wise* $\ell_\infty$ *bound* $\|\widehat{B} - B^*\|_\infty \leq C_5 \sqrt{\frac{p^{\tau/k}}{n}}$, *and*

(c) $\widehat{B}$ *satisfies sign consistency if* $|B_{\min}^*| \geq 2C_5 \sqrt{\frac{p^{\tau/k}}{n}}$,

*where* $C_4 = \left[48(\max_i \Sigma_{ii}^*)(C_k(M_k+1))^{1/2k} \nu_{D^2}\nu_{B^*} \max\{\nu_{\Gamma^{*-1}}\nu_{B^{*-1}}, 2\nu_{\Gamma^{*-1}}^2\nu_{B^{*-1}}^3, 2\alpha^{-1}d^{-1}\}\right]^2$, $C_5 = 16(\max_i \Sigma_{ii}^*)(C_k(M_k+1))^{1/2k} \nu_{\Gamma^{*-1}}\nu_{D^2}\nu_{B^*}$.

*Proof.* The proof follows along the same lines of Theorem 1. Hence, to avoid redundancy, we provide only high-level details. Part (a) We use the polynomial type tail bound in A.7 to compute $n_f(\delta, p^\tau)$, we therefore have $n_f(\delta, p^\tau) = \frac{(c_* p^\tau)^{1/k}}{\delta^2}$ and substituting for $c^*$ and $\delta$ as given in Lemma A.7 and Lemma A.6 respectively, we get

$$n_f(\delta, p^\tau) = C_4 d^2 p^{\tau/k}. \tag{98}$$

Part(b): From Lemma A.7, we have $f(n, \delta) = \frac{n^k \delta^{2k}}{c_*}$, where $c_* = 2^{2k}(\max_i \Sigma_{ii}^*)^{2k} C_k(M_k + 1)$. Thus setting

$$\delta = \delta_f(n, p^\tau) = \left(\frac{c_* p^\tau}{n}\right)^{1/2k} = 2(\max_i \Sigma_{ii}^*)(C_k(M_k+1))^{1/2k}\sqrt{\frac{p^{\tau/k}}{n}}. \tag{99}$$

On the other hand, from Lemma A.6, we have $\|\widehat{B} - B^*\|_\infty \leq 8\nu_{\Gamma^{*-1}}\nu_{D^2}\nu_{B^*}\delta_f(n, p^\tau)$. Thus,

$$\|\widehat{B} - B^*\|_\infty \leq 16(\max_i \Sigma_{ii}^*)(C_k(M_k+1))^{1/2k}\nu_{\Gamma^{*-1}}\nu_{D^2}\nu_{B^*}\sqrt{\frac{p^{\tau/k}}{n}}. \tag{100}$$

Part (c): similar to the contradiction argument in Theorem 1. The details are omitted. □

We present Frobenius and spectral norm consistency results for distributions with bounded moments.

**Corollary 2.** *Suppose the hypotheses in Theorem 2 hold. Then with probability greater than* $1 - \frac{1}{p^{\tau-2}}$:
$\|\widehat{B} - B^*\|_F \leq C_5 \sqrt{\frac{(s+p)(p^{\tau/k})}{n}}$ *and* $\|\widehat{B} - B^*\|_2 \leq C_5 \min\{d, \sqrt{s+p}\}\sqrt{\frac{p^{\tau/k}}{n}}$, *where* $s = |\mathcal{E}(B^*)|$.

*Proof.* The proof follows along the same lines of Corollary 1. Hence, the details are omitted. □