# OpenReview forum: "Learning the Structure of Large Networked Systems Obeying Conservation Laws"
_NeurIPS.cc/2022/Conference — NeurIPS 2022 Accept_

### Official Review · Reviewer_8D2X · 2022-07-08

**Rating:** 6
**Confidence:** 2
**Soundness:** 3 good
**Presentation:** 3 good
**Contribution:** 2 fair

**Summary:**

This paper proposes a $\ell^1$-regularized log-determinant program for estimating the graph structure encoded in a matrix $B$ from a conservation law $BX=Y$, where the node potentials $Y$ are measured and the injected signals $X$ are Gaussian.
Given the sample covariance of $Y$, the objective function is recast into a $\ell^1$-regularized log-determinant program on $B$. The new objective function is shown to be convex with a unique minimizer $\hat B$.
Furthermore, under relatively standard assumptions, the solution $\hat B$ is shown to match several properties of the actual graph matrix.
Some experiments are presented to show the practical benefits of the proposed optimization problem

**Questions:**

I find Figure 1 difficult to understand. Why do (c) and (d) have a missing edge? In what way are the spurious edges removed in (d)? I believe the presentation would improve significantly with a more detailed discussion here on the related work and approaches.

L65-71: This example is not clear. The equation $B^*=(\sum_l h_l A^L)^{-1}$ seems odd as $A^L$ does not depend on $l$. Also what does it mean that $L=3$ and $B^*=(I-\alpha A)^{-1}$ are reasonable choices? $B^*$ is a function of $L$ and this function does not coincide with $(I-\alpha A)^{-1}$ when $L=3$. Finally, I believe that in most cases the matrix $B^*=(I-\alpha A)^{-1}$ is almost full, so it does not seem like a good example of application of your approach

The experiments only consider very small graphs (up to 64 nodes) but this is claimed to be a method for high-dimensional networks. I find this a little odd, please explain.

Only a very small set of (three) graphs is considered in the experiments. The paper would highly benefit from a more extensive experimental section where a larger number of graphs with different structural properties is considered.

In the experiments it should be highlighted whether or not the assumptions of Thm 1 are satisfied.

In the synthetic experiments $B$ is chosen as the adjacency matrix of the graph, and it is then perturbed to become pd. More details on the precise perturbation made are needed here.

L361: The empirical support recovery probability should be defined more precisely

Figure 3 should report vertical lines to highlight the value of $d^2\log p$ and $d^4\log p$.

The IEEE 33 network is not regular, so what is $d$?

The experiments only verify the sparsity structure. How about sign consistency and norm distance $\|\hat B-B^*\|$?

It is not clear to me whether your derivations and approach work also for weighted (non-binary) graphs.

L121: $T_1,T_2$ are subsets of $[p]$ not $[p]\times [p]$.
L125: the sentence is not clear
L:135: $\Theta^*$ requires some inverse


**Limitations:**

It seems like one limitation of the approach is that it works for relatively small graphs only. This should be clarified in the paper.
Also, it is not clear whether the assumptions of Theorem 1 and its corollaries are verified in practice and how stringent they are. This should be discussed in more details and verified on the graphs the method is tested on.

**Strengths And Weaknesses:**

The paper well written and the contribution is well presented. The whole theoretical machinery quite tightly follows previous work on high-dimensional covariance estimation in [39], the main difference being the fact that the objective function is quadratic in the variable $B$. While this certainly complicates the analysis with respect to [39], the novelty with respect to [39] from the theoretical point of view may be limited.

At the same time, when compared with previous GLASSO technique(s) for recovering the graph structure based on the same $\ell^1$-regularized log-determinant program (e.g. [54,16,13]), the proposed approach provides interesting novelties. In particular, while GLASSO allows to reconstruct the graph pattern only for paths of length two, and thus requires $O(d^4\log p)$ samples to reconstruct the graph, the proposed estimator recovers the graph with only $O(d^2\log p)$ samples.

The statements of the main theorems/results seem correct and reasonable, although I did not check the proofs in details.

---

> ### Author Response · Authors · 2022-08-02
> **Caption of Figure 1 and typos in some expressions of the brain networks motivating example**
>
> **Figure 1 caption**: Fig.1 is a "stylistic" or "conceptual" visualization of how our $\ell_{1}-$regularized MLE and the GLASSO-based approach differ in terms of recovering the true structure of the unknown network. Specifically, in Fig 1(c) and 1 (d), we depict potential outputs (random objects) of $\ell_{1}-$regularized MLE and GLASSO+2HR for some sample covariance matrix. Since the outputs are random objects, the graphs in Fig 1 (c) and 1 (d) can have missing and false edges that are not present in the true graph in Fig 1 (a). However, since we directly estimate sparse $B^{\ast}$, likely, the estimated and the true graph might only differ on a few edges, as shown in Fig 1 (c). Instead, the GLASSO+2HR method advocated in [13] recovers $B^{\ast}$ by first estimating $\Theta^{} = B^{\ast}{}\Sigma_{X}B^{\ast}{}$ that is potentially dense compared to $B^{\ast}$. This is apparent by comparing graphs in Fig 1(a) and 1(b). Hence, the GLASSO+2HR estimate might have more edges than the true graph, as shown in Fig 1 (d). Our experimental results in Section 4 validate our claim. We thank the reviewer for pointing out this source of confusion, and we will elaborate on this explanation in the revised manuscript.
>
> **Q2: L65-71: This example is not clear. The equation $B^{\ast}{}=(\sum_{l=0}^Lh_{l}A^{L})^{-1}$ seems odd as $A^{L}$ does not depend on $l$. Also what does it mean that $L=3$ and $B^{\ast}{} = (1-\alpha A)^{-1}$ are reasonable choices? $B^{\ast}{}$ is a function of $L$ and this function does not coincide with $(1-\alpha A)^{-1}$ when $L=3$. Finally, I believe that in most cases the matrix $B^{\ast}{} = (1-\alpha A)^{-1}$ is almost full, so it does not seem like a good example of application of your approach}**
>
> Thanks for pointing out inconsistencies in this example. There are two typographical errors in the expressions for $B^{\ast}$.
>
> 1. $A^L$ should be $A^l$, and $l=\{0,\ldots,L-1\}$. Thus, $B^{\ast}{}=(\sum_{l=0}^{L-1}h_{l}A^{l})^{-1}$ is the right term.
> 2.  $B^{\ast}=(I-\alpha A)^{-1}$ should be $B^{\ast}=(I-\alpha A)$; that is, there is no inverse sign.
>
>  By $L=3$ and $B^{\ast}=(I-\alpha A)$ (the corrected expression) are reasonable choices we mean the following: in Brain applications [27, 42], the typical choices of the graph filters are $B^{\ast}=(\sum_{l=0}^2h_lA^l)^{-1}$ and $B^{\ast}=(I-\alpha A)$. In the former, we let $L=3$. Our previous write-up might have implied that we are substituting $L=3$ in $B^{\ast}{}=(\sum_{l=0}^{L-1}h_{l}A^{l})^{-1}$ to obtain $B^{\ast}=(I-\alpha A)$. This is incorrect. We will correct the typos in the revised manuscript and re-write the sentences to enhance clarity.
>
>  Finally, given the correct expression of $B^{\ast}=(I-\alpha A)$, note that $B^{\ast}$ is sparse when $A$ is sparse; and hence, our approach is applicable.

---

> ### Author Response · Authors · 2022-08-02
> **Clarification on experiments and experimental setup.**
>
> **The experiments only consider very small graphs (up to 64 nodes) and only a very small set of (three) graphs**
> **A:** Our theoretical results demonstrate that the proposed estimator $\widehat{B}$ is a high dimensional convex estimator or in other words $\widehat{B}$ is a good estimate of $B^{\ast}{}$ even in situations where the number of samples $n$ is (significantly) smaller than the number of variables $p$. The precise characterization of the theoretical properties of the proposed algorithm for this important problem is the main focus of the paper. The experiments are designed to be proof-of-concept and we believe they demonstrate strong support for the theory. That said, the proposed estimator may be improved from the viewpoint of computational efficiency and this would be key to carrying out extensive evaluations on large scale problems. It is worth noting that the chain and grid graphs are two standard graph models usually considered in literature as proof-of-concepts and many realistic graphs such as the IEEE power distribution networks may be realized as a combination of these two models. We think that a large scale empirical study would indeed be interesting and this is an exciting avenue for future work.
>
> **In the experiments, it should be highlighted whether or not the assumptions of Thm 1 are satisfied**
>
> **A**  In practice, the proposed algorithm/estimator is not required to verify the assumptions [A1-A3] mentioned in the paper, because it recovers the unknown structure of the network by finding the unique minima of the $\ell_{1}-$regularized convex program defined in equation (3). Notice that from Lemma 1 the loss function/objective is convex and admits a unique solution if the regularization constant $\lambda_{n}>0$ and $B$ is positive definite, therefore for the purposes of simulation as long as $\lambda_{n}>0$ and $B\succ 0$ verifying assumptions [A1-A3] are not necessary for the algorithm to recover the unique minima.
> On the other hand, to theoretically guarantee the strong properties of the estimator, such as support recovery, $\ell_{\infty}$ norm consistency, and sign consistency, we need to make assumptions [A1-A3] on the ground truth model. We note that such assumptions are standard in the graphical model selection and high-dimensional statistics literature and may be somewhat conservative in real-world problems. Indeed, one may interpret the compelling experimental results as validation for the ability of the proposed algorithm to perform well in realistic situations that may very well be outside the constraints of our theory.
>
> We would also like to add that there is an active line (see for ref [1-5]) of research devoted to testing whether such assumptions are satisfied by setting up a semidefinite program to estimate the parameters underlying the assumption. It will indeed be interesting to explore such questions in the context of the present problem as part of future work, and we will add a discussion to this effect in the revised version. Thanks for the interesting question!
>
> (1) d’Aspremont, Alexandre, and Laurent El Ghaoui. ``Testing the nullspace property using semidefinite programming.” Mathematical programming 127.1 (2011): 123-144
>
> (2) Koiran, Pascal, and Anastasios Zouzias. ``Hidden cliques and the certification of the restricted isometry property.“IEEE transactions on information theory 60.8 (2014): 4999-5006.
>
> (3) Tang, Gongguo, and Arye Nehorai. ``Fixed point theory and semidefinite programming for computable performance analysis of block-sparsity recovery.” arXiv preprint arXiv:1110.1078 (2011).
>
> (4) Wang, Tengyao, Quentin Berthet, and Yaniv Plan. ``Average-case hardness of RIP certification.” Advances in Neural Information Processing Systems 29 (2016).
>
> (5) Bandeira, Afonso S., et al. ``Certifying the restricted isometry property is hard.” IEEE transactions on information theory 59.6 (2013): 3448-3450.

---

> ### Author Response · Authors · 2022-08-02
> **Clarifications on experiments and experimental setup**
>
> **In the synthetic experiments $\mathbf{B}$ is chosen as the adjacency matrix of the graph, and it is then perturbed to become pd. More details on the precise perturbation made are needed here.**
>
> **A:** The perturbation is of the form $B^{\ast}{} = B+c.I$, where $B$ is the adjacency matrix, $I$ is the identity matrix and $c$ is any constant which makes $B^{\ast}{}$ positive definite. We set $c=2$ in our simulations, but the constant $c$ can be anything as long as $B^{\ast}{}$ is guaranteed to be positive definite. We will expand on this in our revised manuscript.
>
> **The empirical support recovery probability should be defined more precisely, report vertical lines to highlight the value of $\mathbf{d^{2}\log p}$ and $\mathbf{d^{4}\log p}$ and experiments to verify norm and sign consistency.**
>
> **A:** We appreciate the reviewer's valuable suggestions for improving the quality of experimental results. For the purposes of more clarity we describe here the notion of empirical support recovery (line 343) used in experiments, we draw 100 batches of $n$ samples from the distribution and calculate the empirical probability of success as the fraction of 100 batches where the sparsity pattern of the estimator coincides exactly with that of the true matrix $B^{\ast}{}$. We agree that providing more information in terms of adding plots to verify norm and sign consistency to support theoretical results would enhance the paper, and this will be addressed with updates in the revised version.
>
> **The IEEE 33 network is not regular, so what is $\mathbf{d}$?**
>
> **A:** We denote by $d$ the {\em maximum degree} of the network, therefore in the case of the IEEE 33 bus power distribution network, the maximum degree $d=4$. Furthermore, we emphasize that our methods consistently work for any general non-regular graph and do not presume that the underlying graph is regular.
>
> **It is not clear to me whether your derivations and approach work also for weighted (non-binary) graphs.**
>
> **A:** We do not assume that $B$ to be binary and the proofs for Theorem 1 and Theorem 2 hold true for any symmetric positive definite matrix $B^{\ast}{}$. Therefore the results hold for any symmetric positive definite non-binary matrix matrix $B^{\ast}{}$. We will include a remark in the revised version for better clarity.
>
> **L121: $T_1,T_2$ are subsets of $[p]$ not $[p]\times [p]$ ,L125: the sentence is not clear L:135: $\Theta^{}$ requires some inverse**
>
> **A:** We appreciate the reviewer's careful and in-depth reading and we will fix the minor typos and provide more clarification on the sections mentioned.

---

> ### Author Response · Authors · 2022-08-07
> **Followup on author response**
>
> Dear reviewer,
>
> We would like to touch base with you to see whether you had a chance to look at our response. We hope that it has helped address the concerns you have raised in your review. If there are other concerns or if you have more questions, we will be more than happy to provide additional clarification.
>
> Thanks again for your valuable time!
>
> Best,
>
> Authors

---

> > ### Comment · Reviewer_8D2X · 2022-08-08
> > **Thanks**
> >
> > Dear author(s), thanks a lot for your detailed responses, I am quite happy with the way you addressed the issues I have pointed out.

---

> > > ### Author Response · Authors · 2022-08-09
> > > **Thank you!**
> > >
> > > We are glad that our responses helped clarify things. We are also grateful for your careful reading of the paper and for your valuable feedback. This will certainly allow us to refine the quality of our manuscript.
> > > We wonder if you think our responses and planned updates warrant a revision of your initial score for the paper.

---

### Official Review · Reviewer_fKZ9 · 2022-07-12

**Rating:** 7
**Confidence:** 2
**Soundness:** 3 good
**Presentation:** 4 excellent
**Contribution:** 3 good

**Summary:**

The paper developed a $\ell_1$-regularized MLE of the sparsity structure of $B^∗$ in the conservation laws $X=B^* Y$ observed in many networked systems. The authors assumed that the covariance matrix $\Sigma_X$ of the node injections $X$ is known, and the proposed MLE can recover the topology of the structure of the graph from $B^*$ with high probability. With appropriate choices of the regularization coefficient $\lambda_n$ and the sample size $n\ll p$, the proposed estimator is theoretically sound and proved to be effective on synthetic datasets and a real power distribution network. The authors provided detailed proofs to reach the conclusions on the support recovery performance. Also, they compared the proposed MLE against the baselines GLASSOR+2HR and GLASSOR+SR, and the proposed MLE outperformed.


**Questions:**

Minors:

1. Last sentence in the caption of Fig1: ... there are no **spuruious** edges in it -> ... there are no **spurious** edges in it.
2. L125: $\lVert A \lVert_{\infty}$ has two more vertical lines?
4. L187: ...might not be be strictly convex. -> ...might not be strictly convex.
5. An explicit definition of the support of $B^*$ will be better.

**Limitations:**

YES.

**Strengths And Weaknesses:**

Strengths:

1. The paper is well-written and developed a $\ell_1$-regularized MLE for the support of $B^*$ from limited potential samples.
2. The proposed estimator is proved to be convex in B, therefore, it has a unique solution regardless of the size of problem.
3. It is consistent to the true $B^*$ in terms of elementwise maximum, Frobenius and operator norms.
4. Theorems of the support recovery are given for Gaussian, sub-Gaussian node potentials and the ones with bounded moments.
4. Relatively, it requires less number of potentials samples than the baseline GLASSOR+SR.

Weaknesses:

The developed estimator claimed to be able to avoid the assumptions that are imposed on prior works, such as triangle-freeness of $\mathcal G$, no correlation between the node injections, and the invertibility of the empirical covariance matrix. However, it relies on three necessary assumptions [A1,A2 and A3]. How to verify that these conditions hold in practice?

---

> ### Author Response · Authors · 2022-08-02
> **Response to Reviewer fKZ9**
>
> **Q1: The results rely on three necessary assumptions [A1,A2 and A3]. How to verify that these conditions hold in practice?**
>
> **A1:** It is worth noting that in order to run our proposed algorithm/estimator, one need not verify the assumptions. Indeed it recovers the unknown structure of the network by finding the unique minima of the $\ell_{1}-$regularized convex program defined in equation [3]. Notice that from Lemma 1 the loss function/objective is convex and admits a unique solution if the regularization constant $\lambda_{n}>0$ and $B$ is positive definite.
>
>  On the other hand, in order to theoretically guarantee the strong properties of the estimator, such as support recovery, $\ell_{\infty}$ norm consistency, and sign consistency, we need to make assumptions [A1-A3] on the ground truth model. We note that such assumptions are standard in the graphical model selection and high-dimensional statistics literature.
>
>   We would also like to add that that there is an active line (see for ref [1-5]) of research devoted to testing whether such assumptions are satisfied in the problem at hand by setting up a semidefinite program to estimate the parameters underlying the assumptions. It will indeed be interesting to explore such questions in the context of the present problem as part of future work  and we will add a discussion to this effect in the revised version. Thanks for the interesting question!
>
> 1)d’Aspremont, Alexandre, and Laurent El Ghaoui. ``Testing the nullspace property using semidefinite programming." Mathematical programming 127.1 (2011): 123-144.
>
> (2) Koiran, Pascal, and Anastasios Zouzias. ``Hidden cliques and the certification of the restricted isometry property."IEEE transactions on information theory 60.8 (2014): 4999-5006.
>
> (3) Tang, Gongguo, and Arye Nehorai. ``Fixed point theory and semidefinite programming for computable performance analysis of block-sparsity recovery." arXiv preprint arXiv:1110.1078 (2011).
>
> (4) Wang, Tengyao, Quentin Berthet, and Yaniv Plan. ``Average-case hardness of RIP certification." Advances in Neural Information Processing Systems 29 (2016).
>
> (5) Bandeira, Afonso S., et al. ``Certifying the restricted isometry property is hard." IEEE transactions on information theory 59.6 (2013): 3448-3450.
>
> **Q2:  An explicit definition of the support of $B^{*}$ will be better, comment regarding minor typos.**
>
> **A2:** Thanks for the helpful suggestion! To clarify, the augmented set $E$ is the support of $B^{\ast}$, as defined in L204 the set $E = \{\mathcal{E}(B^{\ast})\cup (1,1)\ldots\cup (p,p)\}$, where $\mathcal{E}(B^{\ast}) = \{(i,j)\in [p]\times [p]: B^{\ast}_{ij}\neq 0, \text{for all} \hspace{3px} i\neq j\}$ is the edge set of the network $\mathcal{G}$, however for the purposes of better readability we will include a more concise definition for the support and rectify the minor typos in the revised version.

---

> > ### Author Response · Authors · 2022-08-07
> > **Followup on author response**
> >
> > Dear reviewer,
> >
> > We would like to touch base with you to see whether you had a chance to look at our response. We hope that it has helped address the concerns you have raised in your review. If there are other concerns or if you have more questions, we will be more than happy to provide additional clarification.
> >
> > Thanks again for your valuable time!
> >
> > Best,
> >
> > Authors

---

### Official Review · Reviewer_tryY · 2022-07-14

**Rating:** 7
**Confidence:** 2
**Soundness:** 3 good
**Presentation:** 3 good
**Contribution:** 3 good

**Summary:**

The paper deals with the estimation of graphical models under the constraints of conservation laws of network flow. The paper focuses on the problem of estimation of Laplacian matrices using the log-likelihood of multivariate Gaussian distributions. The paper considers solving an optimization problem involving penalized log-likelihood of Gaussian distribution extending the graphical lasso methods. The paper also provides theoretical results on the consistency of the estimated Laplacian matrices as well as support recovery under sparsity conditions. The paper provides both theoretical conditions on the error bounds on the estimates as well as a simulation study demonstrating the effectiveness of the penalized estimation.

**Questions:**

1. Is it possible to address the situation of unknown covariance matrices (which is pretty natural in real data contexts) in more detail?
2. What are the computational costs of these methods and up to what size of networks can be handled by this method?

**Strengths And Weaknesses:**

Strength:
(1) Formulation of a convex problem using penalized log-likelihood for estimation of Laplacian matrices under the conservation laws of the network flow.
(2) Theoretical results on the error bounds on estimated Laplacian using $\ell_\infty$, Frobenius, and operator norms.
(3) Theoretical results on the support recovery of Laplacian matrices under the condition of sparsity in the Laplacian population matrix.

Weakness:
(1) The situation of the unknown covariance matrix in Remark 1 requires more explanations. Rigorous statements on how the problem change in this case, and how the estimation of $B^*$ can still be possible should be mentioned in more detail.
(1) The innovations in the proof techniques for proving Theorems 1 and 2 have not been properly explained. The assumptions of [A1]-[A3], as well as the primal-dual witness construction, are standard in the literature, so a bit more explanation on the difficulty and innovations in section 3.3 would be appealing.

---

> ### Author Response · Authors · 2022-08-02
> **Response to Reviewer tryYPart 2**
>
> **Q2: Is it possible to address the situation of unknown covariance matrices (which is pretty natural in real data contexts) in more detail?**
>
> **A2:** Thank you for the suggestion. We highlight that our problem setup is motivated by applications (see our list in the introduction) where the correlations ($\Sigma_X$) of the injected flows are available. Specific applications include power, water, and transportation networks in which injected flows are periodically measured for running diagnostics or inferred using prior model knowledge. Interestingly, our method is applicable for an unknown $\Sigma_X$, provided it has some specialized structure. To see this, consider the inverse covariance matrix $B^{\ast}\Sigma_{X}^{-1}B^{\ast}=(B^{\ast}D)(B^{\ast}D)^T$, where $\Sigma^{-1}_X=D^2$ is unknown. By imposing an $\ell_1$ penalty on $BD$,  similar to Eq (3), we can cast an $\ell_1$-regularized MLE to estimate $B^{\ast}D$ instead of $B^{\ast}$. As pointed in Remark 1, this estimator help us to estimate the support of $B^{\ast}$ if the sparsity of $B^{\ast}$ (approximately) equals the sparsity of $B^{\ast}D$. For concreteness, let us consider two exact cases: (1) For a diagonal matrix $D$, obviously the sparsity patterns of $B^{\ast}D$ and $B^{\ast}$ coincide. (2) Consider the following Hermitian block matrix, which is typically used in the AC power networks [1]:
>
> $B = [B1,B2; B3, B4]$
>
> where the blocks $B_1$ and $B_2$ has identical support. In this case, the sparsity pattern of $B^{\ast}D$ and $B^{\ast}$ coincide for a diagonal $D$ and also $2\times 2$ block matrix $D$ with each block being a diagonal. Finally, consider an unknown non-diagonal $D$. From the optimization problem in Eq (3), note that there are two unknown variables; that is $D$ and $B^{\ast}$. This renders the $\ell_1$-regularized MLE to be a non-convex, which is out of the scope of our current work and a fruitful avenue for the future research. In the revised manuscript, we will expand on these points in both Remark 1 and in the conclusion.
>
> **Q3: What are the computational costs of these methods and up to what size of networks can be handled by this method?**
>
> **A3:** As is well known the Graphical LASSO has a worst case per iteration complexity of $\mathcal{O}(p^{3})$, where $p$ is the dimensionality of the problem. Since the proposed $\ell_{1}$ regularized MLE is a semidefinite program, it is well known that SDP’s can be solved in polynomial time by well known methods such as interior point method, however the worst case complexity can be vastly improved if the matrix is very sparse, for instance chain graphs. We wish to emphasize that this paper is the first of it’s kind to provide strong theoretical guarantees on the structure recovery property for networked systems obeying conservation laws, however we recognize addressing  the issue of computational complexity of the proposed work is an interesting avenue for future work. We aim to develop techniques along the lines of an active area of research (see for ref [1-4]) where methods are developed to significantly reduce the complexity thus allowing for inference in large scale networks
>
> (1) Fattahi, Salar, Richard Y. Zhang, and Somayeh Sojoudi. ``Sparse inverse covariance estimation for chordal structures." 2018 European Control Conference (ECC). IEEE, 2018.
>
> (2) Friedman, Jerome, Trevor Hastie, and Robert Tibshirani. ``Sparse inverse covariance estimation with the graphical lasso." Biostatistics 9.3 (2008): 432-441.
>
> (3) Mazumder, Rahul, and Trevor Hastie. ``Exact covariance thresholding into connected components for large-scale graphical lasso." The Journal of Machine Learning Research 13.1 (2012): 781-794.
>
> (4) Sojoudi, Somayeh. ``Equivalence of graphical lasso and thresholding for sparse graphs." The Journal of Machine Learning Research 17.1 (2016): 3943-3963.

---

> ### Author Response · Authors · 2022-08-02
> **Response to Reviewer tryY Part 1**
>
> **Q1: Innovation and difficulty in proving the main theorems.**
>
> **A1:** We have highlighted the technical difficulty and differences in several places in the manuscript including in L146-147,196-199,299-301. To summarize,
> - The primary distinction from the Graphical LASSO formulation is that the objective function in the MLE is quadratic in the variable $B$, therefore as mentioned in L181-189, determining whether the objective function was convex and admitted a unique solution was crucial and non-trivial given that the quantities $D$ and $B$ were both quadratic in the trace functional, where $D$ is the unique positive definite square root of $\Sigma_{X}^{-1}$. This gave rise to a different set of necessary assumptions for identifiability and recoverability that are unique to our problem.
> - The known deterministic term $D$ and the unknown variable $B$ are coupled under the trace functional, furthermore this coupling cannot be broken and manifests itself in a bunch of hessian terms making the  the primal dual witness analysis more complex and involved.  This coupling is unique to our problem and the major innovation lies in handling the coupled terms appropriately. This is demonstrated in the proofs of all the auxiliary lemmas, including the sufficient condition for strict dual feasibility and the lemma for controlling the distortion factor $\Delta$.
>
> However we agree with the reviewer that an explanation on the novelty and innovation would be more appealing and enhance readability, towards this we will include an elaborate remark along the same lines in the revised version of the paper.

---

> ### Author Response · Authors · 2022-08-07
> **Followup on author response**
>
> Dear reviewer,
>
> We would like to touch base with you to see whether you had a chance to look at our response. We hope that it has helped address the concerns you have raised in your review. If there are other concerns or if you have more questions, we will be more than happy to provide additional clarification.
>
> Thanks again for your valuable time!
>
> Best,
>
> Authors

---

### Official Review · Reviewer_HeKM · 2022-07-15

**Rating:** 4
**Confidence:** 4
**Soundness:** 3 good
**Presentation:** 3 good
**Contribution:** 2 fair

**Summary:**

This submission studies how to learn the structure of large network that obeys conservation law, a l1-regularized formulation is introduced, and the problem is convex.

**Questions:**

1.what is the exact novelty and challenge compared with GLASSO?

It seems to the reviewer, and also confirmed by the authors that the formulation in (2) is the same as GLASSO, the only difference is B instead of Theta is being estimated. However, this makes the problem convex, and henceforth it seems to be simpler than GLASSO, Given that the problem formulation is simpler and the solving is easier, the novelty and challenge need to be further discussed.

2. the experiments only show synthetic and real power grids data, are there any other application domains? if so, should be discussed and more experimental results in other applications should be include to show the generality of the proposed framework. If not, then the applicability of the framework is quite limited.

**Limitations:**

lack of discussion about the novelty
applicability of the proposed framework seems to be limited to power grids.
societal limitation not discussed

**Strengths And Weaknesses:**

weakness:

1.  the motivation is not well-established.
2.the novelty compared with previous works is not sufficient

---

> ### Author Response · Authors · 2022-08-02
> **Justifying the novelty of the proposed estimator with respect to the traditional GLASSO estimator.**
>
> **Unclear structure and motivation**: Thanks for commenting on the paper's readability. However, we are not sure about what parts of the paper are unclear to the reviewer. Our paper addresses a structure learning problem in large-scale networked systems that obey conservation laws. These laws model interaction among variables in several domains, from physical to financial networks. See the Introduction and the references there for various applications that satisfy these laws. We also highlighted the need for learning structure in many applications, including power, water, and social network. We then cast the structure learning problem as an $\ell_1$-regularized MLE problem, and we discussed similarities and differences between our problem and the GLASSO estimator that estimates the inverse covariance matrix. On the technical side, we also highlighted the need for our approach as the GLASSO based estimator, i.e., GLASSO+2HR, restricts the graph to satisfy other stringent conditions apart from sparsity (see Introduction for more details).
>
> **Novelty and Challenges**: Our log-det program based $\ell_1$-regularized MLE formulation might look "structurally" similar to the GLASSO setup because of the $\ell_1$ penalty. However, the optimization problem, estimation performance, and consistency guarantees are significantly different. We strive our best to stress these points throughout the paper. We summarize them here and will revisit this in our revised version.
>
> Our work has several novel aspects: First, motivated by some important practical applications (see our list in the introduction), we cast the structure learning problem in the network systems as an MLE problem. Our setup is applicable when we have no access to the latent vector (the injected flows), including its correlations in some special cases; see Remark 1 and our answer to Reviewer tryY. Second, our estimation framework is convex for a symmetric $B^\ast$ (Lemma 1). The proof of this statement is not obvious, and our proof and the explanation below Lemma 1 highlight the technical difficulties involved in establishing the result. Third, we extended the vanilla primal-dual proof technique to show several consistent results of our estimator in the high-dimensional regime. This is one of the major technical innovations of our paper. We stressed at multiple places in the paper how we differ from the GLASSO analysis in [39]. We note that several published works (for e.g., ref [55] in our paper)  proved consistency results based on [39], and our work adds to that literature, thereby expanding the utility of the primal-dual witness framework. Fourth, for a fair comparison, we compared and contrasted our scaling laws with the GLASSO estimator (see Remark 3), and we showed that our sample complexity results are much better than that of the GLASSO estimator. We have also verified our theoretical results using extensive numerical simulations, both synthetic and real-world networks.
>
> **Societal limitations**: Thanks for bringing this to our attention. We will add potential societal impact of our work in the revised paper.

---

> > ### Author Response · Authors · 2022-08-07
> > **Follow up on author response**
> >
> > Dear reviewer,
> >
> > We would like to touch base with you to see whether you had a chance to look at our response. We hope that it has helped address the concerns you have raised in your review. If there are other concerns or if you have more questions, we will be more than happy to provide additional clarification.
> >
> > Thanks again for your valuable time!
> >
> > Best,
> >
> > Authors

---

> > > ### Comment · Reviewer_HeKM · 2022-08-08
> > > **response to author rebuttal**
> > >
> > > The reviewer appreciates the response.
> > >
> > > The reviewer also re-read the paper and the reply to the comments of the other reviewers as pointed out by the author.
> > >
> > > The reviewer do agree the formulation is indeed slightly different from GLASSO. Given this fact, the reviewer raised the initial score. However, the other comments regarding the challenge of the problem and the applicability of the method were not addressed.
> > >
> > > Firstly, the authors did not convince the reviewer in terms of the challenge of proving the problem is convex. The reviewer read the comments under Lemma 1, which mentions the difficulty is due to the composition of two convex functions are not necessarily convex, but need to use the notion of a monotone convex function. However, this is well-known. Per Convex optimization by Boyd (https://web.stanford.edu/~boyd/cvxbook/bv_cvxslides.pdf) slides lecture 3, and a quick google search, it is shown that:
> > > Composite function f(x) = h(g(x)) is convex if:
> > >
> > > 1. g convex; h convex nondecreasing
> > >
> > > 2. g concave; h convex nonincreasing
> > >
> > > Hence, the convexity of composition functions is convex and surely needs to use the monotone convex functions. The reviewer did not see the specific difficulty here. Hence, further clarification is needed.
> > >
> > > Secondly, the experiments only focus on power grids, which the reviewer finds insufficient. But this comment was not addressed.

---

> > > > ### Author Response · Authors · 2022-08-08
> > > > **Convexity and applicability to other domains**
> > > >
> > > > Thanks for your comment!
> > > >
> > > > **Regarding convexity**
> > > > You are indeed correct that the recipe for proving the convexity of our objective is similar to the one that you highlighted. But, we must stress that the above argument is only for **scalar functions with scalar domains**, and the recipe is only a sufficient condition, but not a necessary condition. Instead, in our case, the domain of our objective function is the set of real-valued matrices. It is well known that several facts about convexity of scalar functions **completely fail** when one attempts to transport to matrices. There are several issues -- for instance, how does one define monotonicity for matrix functions as there is no complete ordering on the set of matrices? In fact, notwithstanding the above roadblock, our proof of convexity only relies on the symmetry of the matrix (so, we do not even require the natural Loewner partial order to mimic a proof similar to the one shown in Boyd and Vandenberghe's convex optimization book). Indeed generalizing beyond symmetric matrices is highly nontrivial, and we explicitly highlight this as the next steps for our work (see our Remark after the proof of Lemma 1 in the Appendix).
> > > >
> > > > Also, we emphasize that proving the convexity is only the starting point of our main theoretical contributions. Among other things, we explicitly showed using detailed derivations how one can use coercvity arguments and the KKT conditions to recreate a “primal-dual witness style” proof technique that ensures the estimator is unique and it correctly identifies the **unknown** underlying sparse matrix. The ingredients required for this proof are completely different from what corresponding arguments look like for the GLASSO, and we stressed this point in several parts of the paper, including the Appendix (see for instance, initial paragraphs in Section 3.1 and 3.2). In light of these facts, we strongly believe that establishing convexity of our objective function over symmetric matrices and deriving optimal statistical rates is non-trivial and do not follow as a direct corollary to the results in the GLASSO paper [39].
> > > >
> > > > **Experiments and Applicability to Other Domains**
> > > >
> > > > We apologize that we overlooked providing an explicit answer to this concern in our last answer. Thanks for pointing this out.
> > > >
> > > > This paper introduces a novel framework for topology recovery of networked systems that obey conservation laws. In particular, we construct a novel estimator of the topological structure of the network using only coarse statistical information about the node injections and observations of node potentials. We further provide a **precise theoretical characterization** of the proposed estimator and show that it succeeds even when the number of samples $n$ is (significantly) smaller than the number of variables $p$. **This is the main contribution of the work.** The experimental results in our are designed to be proof-of-concept and we believe our synthetic and real experiments demonstrate strong support for our theory.
> > > >
> > > >  Note that we discuss the applicability of our framework for a wide range of problems in the introduction (Lines 56-80) such as brain connectivity inference, structural equation modeling, power networks, linear dynamical (diffusion) networks. In all these cases, we provide extensive support from the literature that our  _data generation_ model has been applied in the past. This provides a _strong motivation_ to study structure recovery problem for such systems, which our paper addresses. Note that it is customary in theoretical papers on structure recovery to restrict their empirical results to only synthetic experiments (such as the chain or grid graphs). We believe our experiments on real power networks go beyond this and provide even stronger support for our theory. We hope that our work can be judged for its theoretical contributions in advancing our understanding of this important practically relevant problem.
> > > >
> > > >  That said, we do agree that a large-scale empirical evaluation of our estimator in all other scenarios would be an interesting avenue for future work. Note that such an evaluation will build on our work, while also focusing on making the algorithm more computationally attractive.
> > > >
> > > > We thank you again for your engagement with the paper. We believe that this discussion will certainly allow us to improve the quality of our manuscript.

---

### Meta-Review · Area_Chair_7mgM · 2022-08-23

**Recommendation:** Accept
**Confidence:** Less certain

**Metareview:**

The reviewers and the area chair judged the paper as technically sound and found that the problem treated is an interesting variant in the spirit of (but different from) the well-known group-LASSO. The range of applicability of the approach was considered promising. While the narrative of the proof leading to a convex problem was judged standard and by itself perhaps less interesting, overall the contribution was judged as valuable to the community and we recommend acceptance of the paper.

**Award:**

No

---

### Decision · Program_Chairs · 2022-09-14

Accept